# Individual face- and house-related eye movement patterns distinctively activate FFA and PPA

Lihui Wang[1,2,3,4], Florian Baumgartner[1], Falko R. Kaule[1], Michael Hanke[5,6] & Stefan Pollmann [1,2,7]*

We investigated if the fusiform face area (FFA) and the parahippocampal place area (PPA) contain a representation of fixation sequences that are typically used when looking at faces or houses. Here, we instructed observers to follow a dot presented on a uniform background. The dot's movements represented gaze paths acquired separately from observers looking at face or house pictures. Even when gaze dispersion differences were controlled, face- and house-associated gaze patterns could be discriminated by fMRI multivariate pattern analysis in FFA and PPA, more so for the current observer's own gazes than for another observer's gaze. The discrimination of the observer's own gaze patterns was not observed in early visual areas (V1 – V4) or superior parietal lobule and frontal eye fields. These findings indicate a link between perception and action—the complex gaze patterns that are used to explore faces and houses—in the FFA and PPA.

[1] Department of Experimental Psychology, Otto-von-Guericke University, Magdeburg, Germany. [2] Center for Behavioral Brain Sciences, Magdeburg, Germany. [3] Institute of Psychology and Behavioral Science, Shanghai Jiao Tong University, Shanghai, China. [4] Shanghai Key Laboratory of Psychotic Disorders, Shanghai Mental Health Center, Shanghai Jiao Tong University School of Medicine, Shanghai, China. [5] Institute of Neuroscience and Medicine, Brain & Behaviour (INM-7), Research Centre Jülich, Jülich, Germany. [6] Institute of Systems Neuroscience, Medical Faculty, Heinrich Heine University Düsseldorf, Düsseldorf, Germany. [7] Beijing Key Laboratory of Learning and Cognition and School of Psychology, Capital Normal University, Beijing, China. *email: stefan.pollmann@ovgu.de

When we look at a face, we carry out distinctive eye movements[1,2] leading to gaze paths that can easily be discriminated from the gaze paths elicited by other objects. In the present study, we investigated whether face- and house-associated gaze paths lead to differential activation in brain areas known to be activated strongly by actually looking at pictures of faces and houses, the fusiform face area (FFA[3]) and the parahippocampal place area (PPA[4]).

Frontal and parietal areas, in particular the frontal and supplementary eye fields and the parietal cortex, support the planning and execution of eye movements. Traditionally, parietal and premotor areas have been seen as intermediaries between the sensory areas, providing the perceptual input for action planning and the (pre-)motor areas that control motor execution. However, perception and action may be more closely interacting than previously thought. Behavioral evidence suggests object representations are linked to appropriate responses (reviewed by Hommel et al. [5]) leading to the concept that object representations bind not only the features of an object into a coherent whole[6] but that they also contain a representation of actions associated with the object[5,7], creating an "event file"[8]. Such an event file should be found in late perception—at the level of object or scene perception rather than at the level of early feature perception—as well as at the level of early action planning—selection of an ecological class of action rather than programming of specific muscular contractions[5].

While there is increasing evidence that neural activity in premotor and motor cortices contains information about perceived stimuli along with information about potential actions (reviewed by Cisek and Kalaska [9]), less is known about processing of actions in perceptual areas of the brain. Viewing pictures of action effectors (e.g. a hand) elicited similar activation than pictures of tools in lateral occipitotemporal cortex (LOTC)[10–12]. Moreover, action may be represented in LOTC in topographical agreement with the body parts involved[13–15]. While most of these studies inferred claims of action representation indirectly from activation patterns elicited by viewing pictures of action effectors, there is also evidence that activation elicited by unseen limb movements and by viewing the body parts involved overlap[15,16].

In motor cortex, the functional topography of motor and premotor areas has likewise been modeled as an overlay of different maps, namely the well-known somatotopic motor homunculus, a map of ethologically relevant action categories and a map of hand location during movements of the arm[17]. If perception and action share a common framework, we may hypothesize that high-level visual cortex may also contain a map of ecological action categories that is tied to the body parts involved in the actions.

In this study, we investigated one specific action category, namely eye movement patterns used when looking at faces—along with a well-investigated control category, namely houses. Eye movements have been found to modulate the activity of early occipital visual areas in past studies. For example, activation changes in area V4 preceded saccade onsets[18] and the pattern of activation in early retinotopic visual areas enabled decoding of eye position[19]. Fixating a position to the left or right of center modulated fMRI activation in several visual areas, even in the absence of other stimuli than the fixation cross[20,21]. Thus, early visual areas contribute to basic processes of eye movement control. However, to the best of our knowledge, representation of eye movement patterns that are tied to specific object categories, like perceiving a face or a house, have not yet been investigated in ventral occipitotemporal cortex, although it contains the FFA and the PPA that have been shown in numerous studies to respond strongly and specifically to faces and houses.

Here, we asked if the activation patterns of the FFA and PPA carry the information to discriminate between fixation sequences that are carried out when we look at a face or a house. At least for face viewing, eye movement patterns are quite stable, even across different viewing environments[22–24]. The general paradigm that we used is to ask participants to follow a sequence of dots with their gaze. The dot sequence replayed a previously recorded fixation sequence during face viewing or house viewing. Importantly, the dot sequences were presented on a blank screen, so that we could investigate activation of the FFA and PPA in the absence of any face or house images.

Face and house-specific gaze tracks can be decoded in the FFA and PPA. Even after removing potential stimulation confounds, self-generated gaze paths can be selectively discriminated in FFA and PPA. We conclude that action patterns are represented in high-level visual cortex, demonstrating a neural basis for close interactions of perception and action.

## Results

**Experiment 1**. In the first experiment, we recorded eye movements in two observers who were looking at faces or houses and extracted the fixation sequence (Fig. 1a). Then we presented a fixation dot following these fixation sequences on a uniform background (i.e. in the absence of any face/house stimuli) to a new group of observers in an fMRI session. As a control condition, inverted face fixation patterns were generated by flipping the upright face-specific fixation sequences along the horizontal axis. Observers did not know that the dots were face or house-associated fixation patterns (or any previously recorded fixation patterns at all). They were instructed to detect subtle changes in the dot color to ensure attentive gaze-following of the dots (Fig. 1b). A training session preceded the scanner session. Accuracy of target detection or correct rejection in a non-target trial was 85.7% (SE 4.8%) in the training session and 95.4% (SE 1.8%) in the scanner-session.

We first analyzed if the fixation sequences conducted while looking at faces or houses differed from each other. For this purpose, we applied a multivariate classification analysis over the spatiotemporal dynamics of the gaze (for exemplary gaze parameters, see Supplementary Table 1). The analysis showed a high discriminability between the gaze pattern of the face versus house conditions (79.8% classification accuracy), face versus pseudo-inverted face (100%) and house versus pseudo-inverted face (96.4%).

Next, we analyzed the correspondence of fixations with dot presentation. The average latency of the eye movements following the jump of the dot to a new location was determined by maximizing the cross-correlation between the eye position and the actual dot position (Fig. 1c). The mean latency was 223.9 ms (SE 3.3 ms) with a mean overall Spearman cross-correlation of rho(spear) = 0.675, indicating that the dot was well followed.

Using a standard localizer for FFA and PPA (see methods), we were able to localize FFA and PPA bilaterally ($t(20) > 2.3$; Table 1 and Fig. 2). Locations of FFA and PPA were consistent with coordinates reported in the literature[25]. FFA showed right hemisphere dominance that may reflect the left visual field superiority for face perception[26]. In order to compare the activation of the category-specific visual areas to category-unspecific areas involved in saccade control, we additionally defined eye-movement-activated ROIs in the superior parietal lobe (SPL) and the frontal eye fields (FEF; Table 1 and Fig. 2). Early retinotopic visual areas (V1, V2, V3, and V4) were examined based on the probability maps in the Juelich Histological Atlas[27] to investigate the specificity of gaze-track representation in high-level visual areas (FFA, PPA; Fig. 2d).

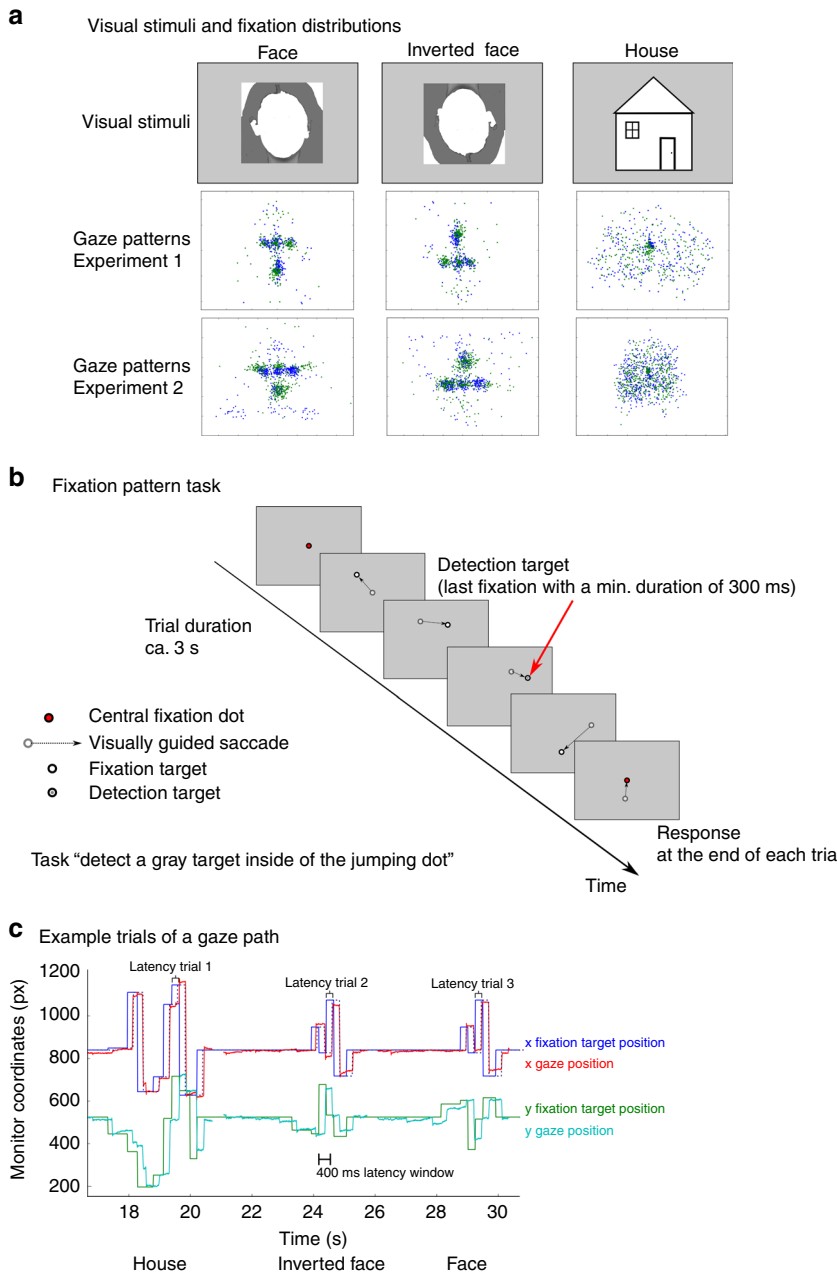

**Fig. 1 Experimental design and fixation patterns in Experiments 1 and 2. a** An example picture for Face, inverted Face (shown as contour here for privacy protection) and a cartoon example of House stimuli (gray-scale pictures were used in the actual experiment; upper panel) and their corresponding fixation patterns which were used for the Fixation Pattern task in Experiment 1 (middle panel) and Experiment 2 (lower panel). The fixation patterns were collected from two subjects (indicated by blue and green, respectively) and pooled over all trials. For Experiment 2, the fixation patterns for houses shown here were normalized to have the same level of spatial dispersion as the fixation patterns for faces (see methods for details). **b** In the Fixation Pattern task the subjects were instructed always to fixate the fixation dot and to indicate at the end of the trial whether they had detected a target. **c** Visualization of the correspondence evaluation between fixation dot position and gaze position of the subject by maximizing the cross-correlation. The position (x, y coordinates, indicated by blue and green, respectively) of the fixation dot and the actual eye position (x, y coordinates, indicated by red and turquoise, respectively) of three consecutive trials collected from one subject are shown as a function of time. The x-axis indicates the time (in seconds) relative to the onset of the first trial. The correspondence was measured by calculating the cross-correlation between the time series of the fixation-dot position and the time series of the actual eye position. Note, the data here is only shown as an example to visualize the two time series.

Pictures of faces and houses typically lead to increased activation in the FFA, respectively, PPA[3,4]. Would this also be the case for face and house-specific gaze tracks? A repeated measures ANOVA on BOLD-signal changes with the factors ROI (PPA, FFA, FEF, SPL) and fixation pattern (Face, inverted Face and House) revealed a significant main effect of fixation pattern ($F_{(2,40)} = 3.72$, $p = 0.033$). House patterns led to stronger activation than face patterns, with inverted face patterns in between. Similarly, the early visual areas (V1–V4) showed stronger activation for house fixation patterns (Fig. 3a, right; see Supplementary Notes for a full ANOVA). The strong activation by house fixation sequences—mirrored by the analogous tendencies in the FEF and SPL—may have been due to low-level differences in saccade amplitude that were higher in

**Table 1 List of regions of interest in 2 mm MNI standard space.**

| Experiment | ROI | Hemisphere | max $t$-value | $x$ | $y$ | $z$ | Cluster size |
|---|---|---|---|---|---|---|---|
| Exp. 1 | FFA | right | 7.59 | 46 | −52 | −22 | 576 |
| | | left | 4.83 | −44 | −54 | −22 | 262 |
| | PPA | right | 13.70 | 30 | −54 | −12 | 2138 |
| | | left | 15.40 | −30 | −50 | −10 | 2150 |
| | SPL | bilateral | 9.56 | −26 | −54 | 50 | 4682 |
| | FEF | right | 7.38 | 28 | −6 | 48 | 896 |
| | | left | 8.94 | −26 | −8 | 50 | 1045 |
| Exp. 2 | FFA | right | 7.10 | 42 | −50 | 18 | 722 |
| | | left | 7.75 | −40 | −46 | −22 | 545 |
| | PPA | right | 12.00 | 26 | −44 | −12 | 1667 |
| | | left | 15.20 | −28 | −52 | −12 | 1521 |
| | SPL | bilateral | 6.68 | 28 | −46 | 42 | 2777 |
| | FEF | right | 6.68 | 36 | 0 | 52 | 813 |
| | | left | 7.84 | −24 | −8 | 46 | 849 |
| Exp. 3 | FFA | right | 6.92 | 42 | −56 | −18 | 1013 |
| | | left | 5.37 | −40 | −50 | −19 | 397 |
| | PPA | right | 19.70 | 18 | −36 | −16 | 763 |
| | | left | 18.2 | −30 | −45 | −12 | 1010 |
| | SPL | bilateral | 9.65 | 46 | −32 | 46 | 5482 |
| | FEF | right | 8.01 | 22 | −10 | 49 | 1310 |
| | | left | 8.72 | −41 | −2 | 49 | 1457 |

*FFA* fusiform face area, *PPA* parahippocampal place area, *SPL* superior parietal lobule, *FEF* frontal eye fields

house than face-specific sequences (Fig. 1a). We will return to this question in Experiment 2.

Our hypothesis that category-specific fixation sequences are represented in the FFA and PPA does not necessarily predict univariate activation differences but rather a differential information content of the activation pattern within a ROI that allows us to discriminate the fixation sequence category. In contrast to the univariate analyses, the classification accuracies exceeded permutation-based chance levels in the FFA and PPA as well as in SPL and FEF (Fig. 4a and Table 2). This was also true on a single-subject level for most subjects (17 out of 21 in the FFA and PPA, and 18 out of 21 in FEF and SPL; Supplementary Fig. 1). We further investigated early retinotopic visual areas and observed that face, house and inverted face gaze tracks were not classified above chance level in areas V1–V4 (Table 3). Moreover, the classification accuracies for Face vs. House tracks were higher in FFA and PPA than in V1–V4, $p < 0.001$ (permutation testing). There were no differences in sensitivity or bias for face or house gaze tracks between ROIs (Supplementary Figs. 2 and 3)

On the one hand, these results confirmed our hypothesis that face and house-specific fixation patterns are represented in FFA and PPA, in the absence of face or house images. The absence of gaze-track discriminability in V1–V4 is in line with the hypothesis that actions are bound to object representations and speaks against a confound of gaze tracks activation patterns with low-level visual features. That gaze tracks could be discriminated not only in the FFA and PPA, but also the SPL and FEF was not unexpected under the hypothesis that high-level visual cortex interacts with visuomotor areas to support action planning.

**Experiment 2.** One important difference between looking at a face and a house is that faces attract more central fixations than houses. This difference in gaze dispersion was left uncorrected in Experiment 1, because it may be an important contributing factor to the representation of face and house-associated gaze-tracks. However, in Experiment 2 gaze dispersion was matched between face and house images, by matching the mean and standard deviation of the house patterns to the face patterns. In most other respects, Experiment 2 was identical to Experiment 1. For the inverted face condition, however, we recorded new fixation

sequences during viewing of inverted faces. This should capture differences in the viewing of upright and inverted faces that might not have been captured by the mere inversion of the fixation sequences for upright faces in Experiment 1.

We predicted that univariate differences in activation elicited by unequal dispersion should be greatly reduced while aspects of face- and house-specific fixation sequences that are not tied to low-level features such as saccade amplitude should still be discriminable in the FFA and PPA.

Dot detection accuracy was 80.7% (SE 2.5%) in the training session and 93.4% (SE 2.1%) in the scanner-session, again confirming attentive task processing. The fixation sequences again were distinct, demonstrated by high classification accuracies (face versus house; 78.6% accurate), face versus inverted face (92.9%) and house versus inverted face (88.1%). In order to investigate if subjects fixated inverted faces in a different manner than upright faces, we re-inverted the gaze patterns recorded during watching inverted faces. The accuracy for the comparison face versus re-inverted face was 79.2%, demonstrating the distinctiveness of inverted gaze patterns beyond inversion. Participants again followed the dots well. The mean latency of eye movements was 254.4 ms (SE 2.5 ms), leading to a mean cross-correlation of rho(spear) = 0.659

In contrast to Experiment 1, neither the main effect of fixation pattern nor the interaction were significant (both $F < 1$; full ANOVA results in Supplementary Notes). Thus, the elimination of dispersion differences between face and house sequences eliminated univariate activation differences between face and house fixation patterns. Nevertheless, the multivariate classification accuracies showed the same pattern as in Experiment 1, exceeding chance levels for all three comparisons in the FFA, PPA, SPL, and FEF, but not in V1–V4 (Fig. 4b and Tables 2 and 3). The classification accuracies for Face vs. House tracks were higher in FFA and PPA than in V1–V4, $p = 0.031$ (permutation testing). Individual accuracies exceeded permutation-based chance levels for 13 out of 18 participants in the FFA and PPA, 14 out of 18 in the FEF and 12 out of 18 in the SPL (Supplementary Fig. 1). Again, there were no differences in sensitivity or bias for face or house gaze tracks between ROIs (Supplementary Figs. 2 and 3).

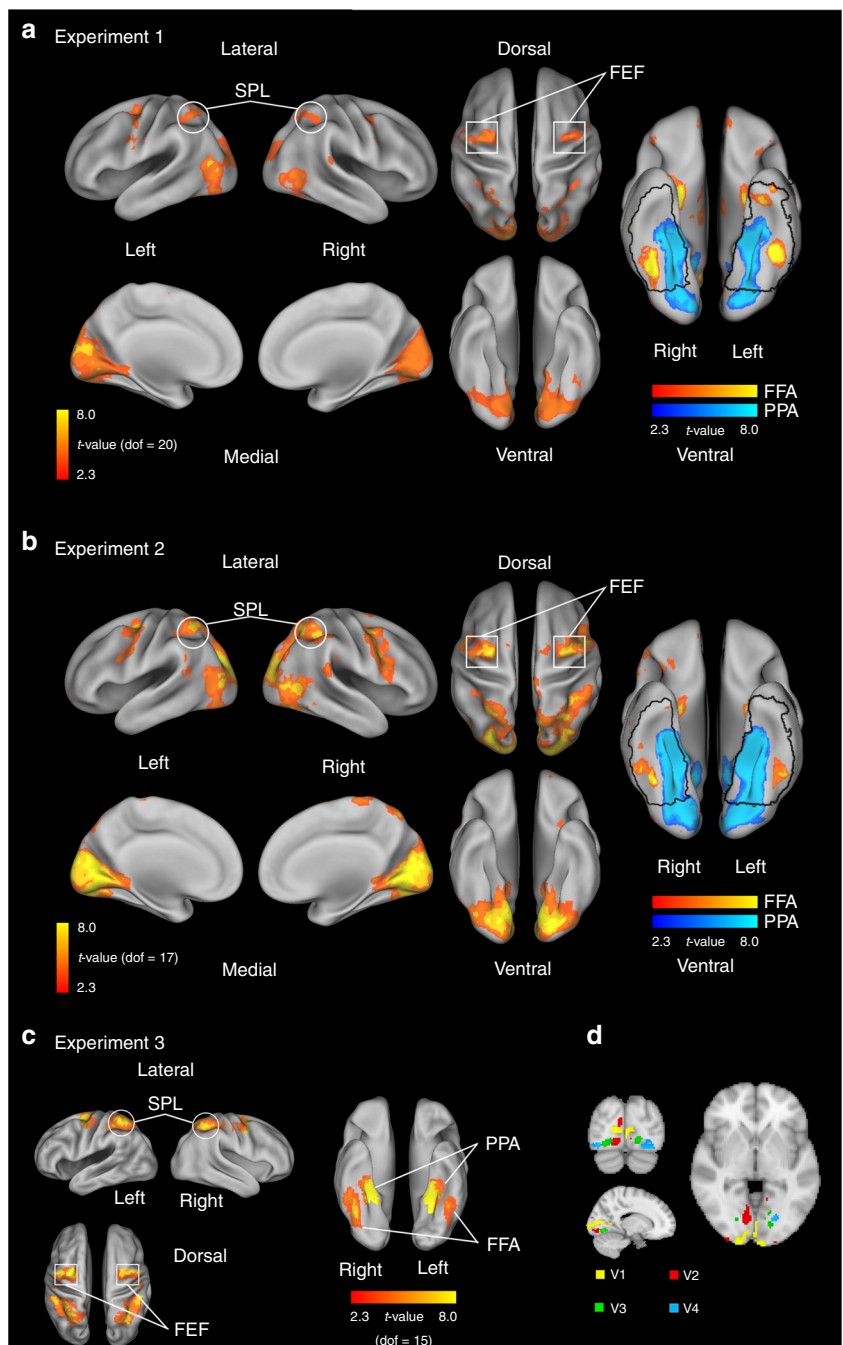

**Fig. 2 Visualization of ROIs. a** Localizer results for Experiment 1. The frontal eye fields (FEF) and the superior parietal lobule (SPL) were identified based on the group-level statistical maps of the fixation pattern contrast Face + House + inverted Face for Experiment 1 (significant on cluster level $p < 0.05$, one-sample $t$ test, cluster forming threshold $z > 2.3$, left). Fusiform face area (FFA) and parahippocampal place area (PPA) were identified based on the visual localizer group-level statistical map of the contrast Face-House ($z > 2.3$; uncorrected, right). The statistical maps are projected onto an inflated brain template using Caret 5.64 with the PALS-B12 atlas[36]. **b** Localization of FEF, SPL (left), FFA and PPA (right) for Experiment 2 (with the same thresholds as Experiment 1). **c** Localization of FEF, SPL (left), FFA and PPA (right) for Experiment 3. FEF and SPL were defined identified based on the group-level statistical maps of the fixation pattern contrast Self-face + Self-house + Other-face + Other-house for Experiment 3 (significant on cluster level $p < 0.05$, one-sample $t$ test, cluster forming threshold $z > 2.3$, left). FFA was identified based on the visual localizer group-level statistical map of the contrast Face > all other pictures (houses, scenes, bodies, objects, scrambled pictures), and PPA based on the contrast (House + Scene) > all other pictures (faces, bodies, objects, scrambled pictures) ($z > 2.3$; uncorrected, right). **d** Localization of early visual areas (V1–V4) were defined by the probability maps in the Juelich Histological Atlas[25].

The replication of the classification results in the absence of univariate activation differences shows that the information content of the activation patterns is independent of overall activation levels. Again, gaze tracks could be discriminated in the two visuomotor ROIs (SPL and FEF) but also in FFA and PPA, again suggesting that FFA and PPA contribute to visuomotor processes.

**Experiment 3**. Experiments 1 and 2 suggest that FFA and PPA support category-specific visuomotor processes. However, is there a specific contribution by FFA and PPA that distinguishes them from other visuomotor areas like SPL and FEF? Experiment 3 investigated this question based on individual differences in face-specific gaze patterns[22,23]. Individual gaze tracks appear to be

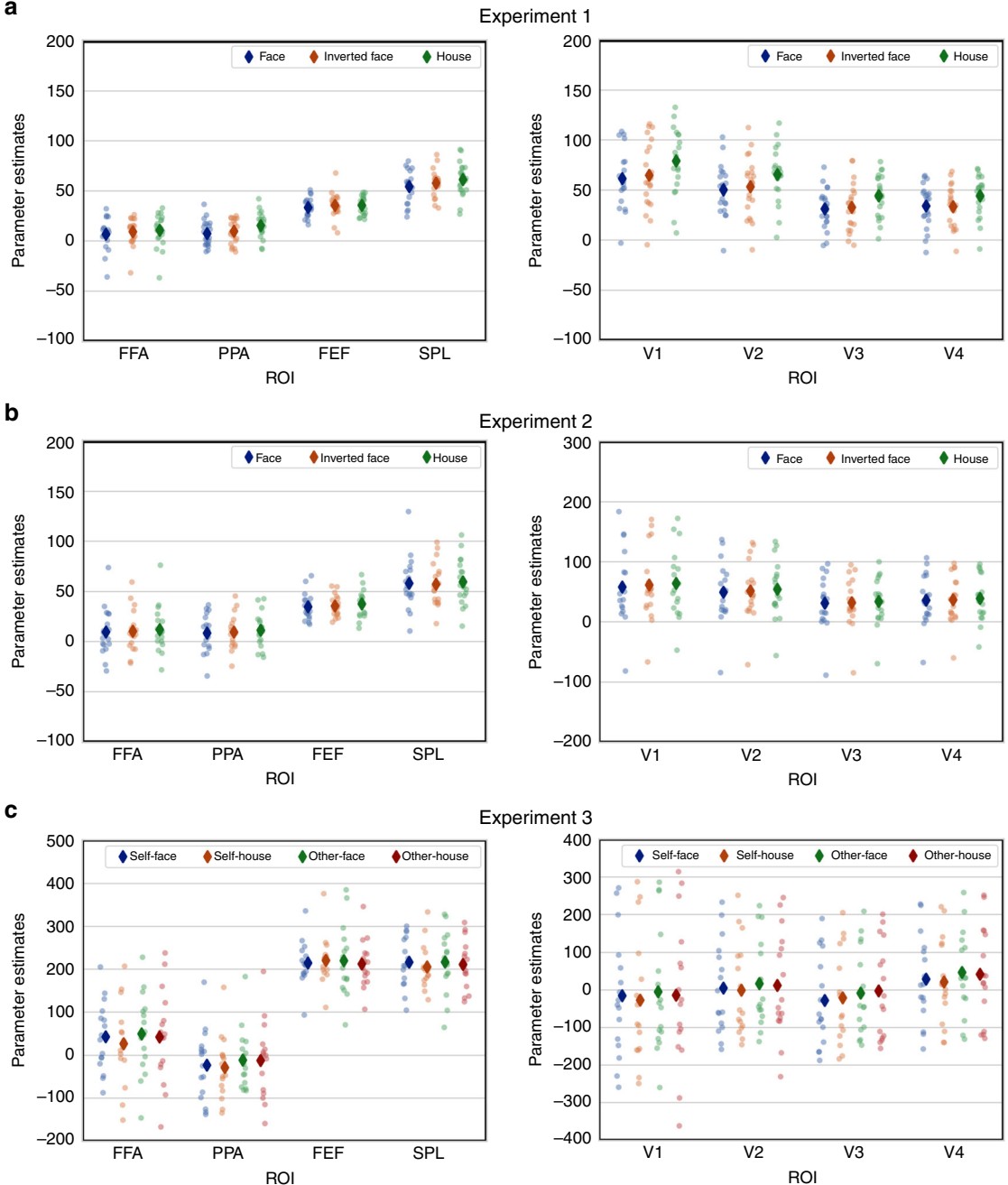

**Fig. 3 Univariate analyses.** Average standardized parameter estimates (diamond) and individual parameters (dots) extracted from the fusiform face area (FFA), parahippocampal place area (PPA), frontal eye fields (FEF), superior parietal lobule (SPL) and early visual areas for the Fixation Pattern conditions for Experiment 1 (**a**) Experiment 2 (**b**), and Experiment 3 (**c**). Individual data under different conditions are shown as color scales (Experiment 1 and 2: blue dots for face, orange dots for inverted face, green dots for house; Experiment 3: blue dots for self-face, orange dots for self-house, green dots for other-face, red dots for other-house).

remarkably stable, even across face viewing under lab and real-world viewing conditions[24]. If these face-viewing patterns are represented in the FFA, then participants' own viewing patterns should be the optimal stimuli to elicit distinct activation patterns in FFA and PPA. In Experiment 3, we recorded fixation sequences during face viewing from the same participants that 1 week later took part in the dot-tracking task (analogous to Experiments 1 and 2) in the scanner (Fig. 5a). Half of the fixation sequences in the scanner were the participant's own fixation sequences, whereas the other half was taken from another participant and vice versa for this participant. We expected that

participants' own fixation sequences for faces and houses would lead to more discriminable activation patterns than other participants' fixation sequences for faces and houses. Importantly, on the group level the comparison of self vs. other gaze tracks relied on the identical set of gaze tracks on both sides of the comparison. Thus, the comparison of self-generated and other fixation sequences allowed us to rule out low-level perceptual or motor processes as confounds. Note that Experiment 3 focused mainly on face processing, because equivalent reports about individual differences of house viewing patterns are (to the best of our knowledge) not available.

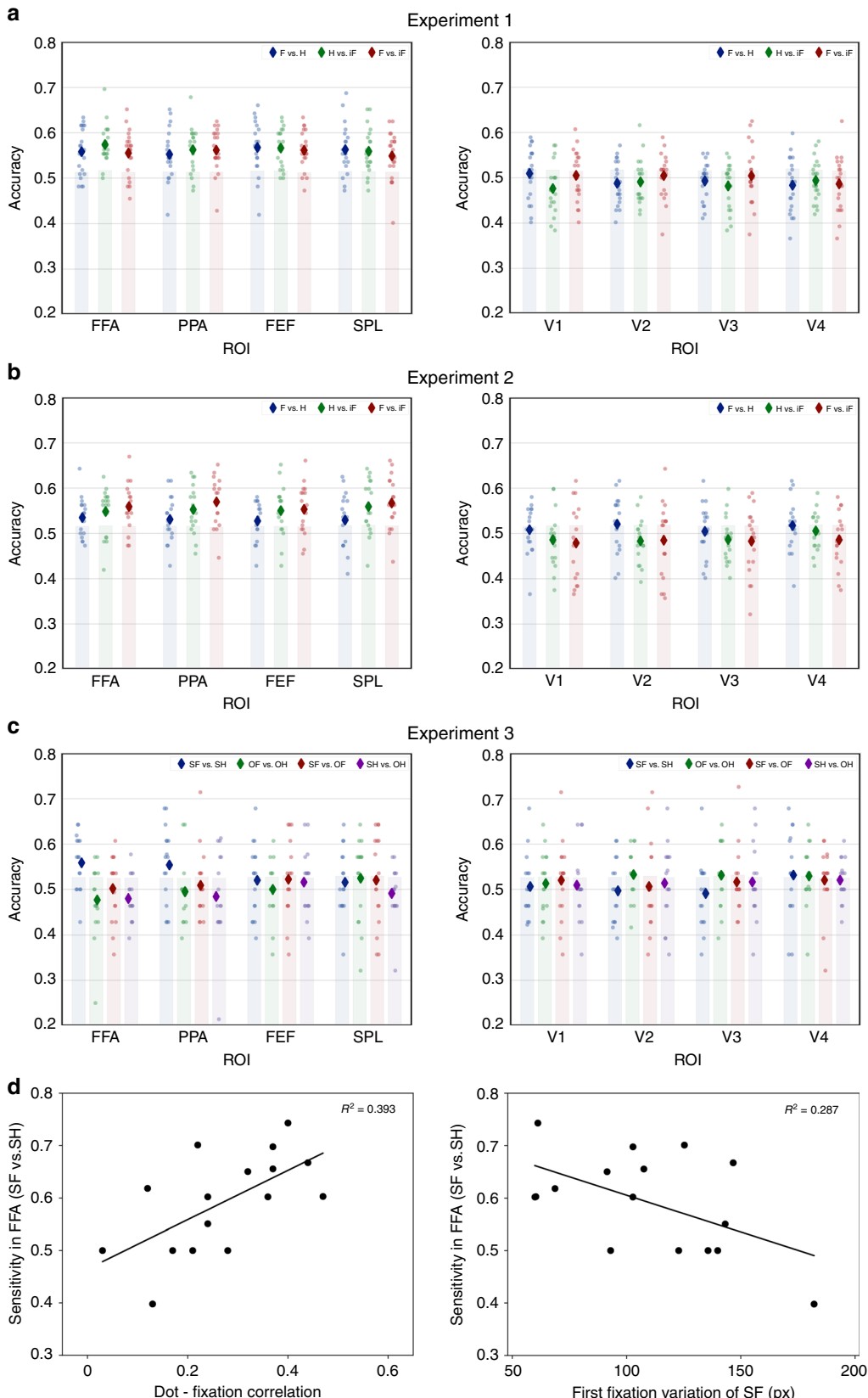

We also shortened the duration of the gaze tracks in an effort to reduce the fixation sequence to its most informative part (Fig. 5c). In a previous study[28], only the first two saccades contributed significantly to face recognition. To capture only these initial saccades, the dot's movement represented only the first 600 ms of the gaze path (vs. 3000 ms in Experiments 1 and 2). In this way, we further eliminated non-essential eye movements that otherwise may have confounded face and house specific fixation behavior.

**Fig. 4 Multivariate analyses. a** Average accuracies (diamonds) and individual accuracies (dots) extracted from the FFA, PPA, FEF, SPL (left panel), early visual areas (V1–V4, right panel) for the pairwise classifications in Experiment 1 (F: face, H: house, iF: inverted face). The shaded areas indicate accuracies below the 95th percentile of the null distribution obtained from the permutations. Individual data under different pairwise classifications are shown as color scales (blue dots for face vs. house, green dots for house vs. inverted face, red dots for face vs. inverted face). **b** The results of the same analysis as (A) in Experiment 2.**c** Average accuracies (diamonds) and individual accuracies (dots) extracted from the FFA, PPA, FEF, SPL (left panel), early visual areas (V1–V4, right panel) for the pairwise classifications in Experiment 3 (SF: self-face, SH: self-house, OF: other-face, OH: other-house). Individual data under different pairwise classifications are shown as color scales (blue dots for self-face vs. self-house, green dots for other-face vs. other-house, red dots for self-face vs. other-face, purple dots for self-house vs. other-house). **d** Scatter plots (with best-fitting regression lines) illustrate the individual sensitivity (area under the curve, see methods and Supplementary Fig. 2) for the classification 'self-face vs. self-house' in FFA as a function of the cross-correlation between the gaze-track position and the actual eye position (left panel), and as a function of the variation of the first fixation locations for self-face (right panel) in Experiment 3. The variation was obtained by calculating the mean distance (in pixel) among the first fixation locations across all trials.

**Table 2 Multivariate decoding results of high-level areas.**

| Experiment | ROI | N voxels | Classification | Mean (%) | SE (%) | p |
|---|---|---|---|---|---|---|
| Exp. 1 | FFA | 679.3 ± 157.2 | Face vs. House | **55.8**[a] | 1.1 | $<10^{-5}$ |
| | | | House vs. inverted Face | **57.4**[a] | 0.9 | $<10^{-5}$ |
| | | | Face vs. inverted Face | **55.5**[a] | 1.1 | $<10^{-5}$ |
| | PPA | 1052.3 ± 232.3 | Face vs. House | **55.2**[a] | 1.2 | $<10^{-5}$ |
| | | | House vs. inverted Face | **56.3**[a] | 1.1 | $<10^{-5}$ |
| | | | Face vs. inverted Face | **56.2**[a] | 1.0 | $<10^{-5}$ |
| | FEF | 853.5 ± 181.0 | Face vs. House | **56.8**[a] | 1.2 | $<10^{-5}$ |
| | | | House vs. inverted Face | **56.6**[a] | 0.9 | $<10^{-5}$ |
| | | | Face vs. inverted Face | **56.1**[a] | 0.9 | $<10^{-5}$ |
| | SPL | 999.9 ± 234.2 | Face vs. House | **56.3**[a] | 1.2 | $<10^{-5}$ |
| | | | House vs. inverted Face | **55.9**[a] | 1.1 | $<10^{-5}$ |
| | | | Face vs. inverted Face | **54.9**[a] | 1.2 | $<10^{-5}$ |
| Exp. 2 | FFA | 831.2 ± 168.1 | Face vs. House | **53.5**[a] | 1.0 | $1.4*10^{-4}$ |
| | | | House vs. inverted Face | **54.8**[a] | 1.2 | $<10^{-5}$ |
| | | | Face vs. inverted Face | **55.9**[a] | 1.2 | $<10^{-5}$ |
| | PPA | 955.4 ± 255.0 | Face vs. House | **53.1**[a] | 1.2 | $4.4*10^{-4}$ |
| | | | House vs. inverted Face | **55.3**[a] | 1.2 | $<10^{-5}$ |
| | | | Face vs. inverted Face | **56.9**[a] | 1.3 | $<10^{-5}$ |
| | FEF | 819.7 ± 217.8 | Face vs. House | **52.7**[a] | 1.0 | $2.0*10^{-3}$ |
| | | | House vs. inverted Face | **55.0**[a] | 1.3 | $<10^{-5}$ |
| | | | Face vs. inverted Face | **55.3**[a] | 1.3 | $<10^{-5}$ |
| | SPL | 912.4 ± 251.4 | Face vs. House | **52.9**[a] | 1.4 | $1.9*10^{-3}$ |
| | | | House vs. inverted Face | **55.9**[a] | 1.4 | $<10^{-5}$ |
| | | | Face vs. inverted Face | **56.7**[a] | 1.4 | $<10^{-5}$ |
| Exp. 3 | FFA | 1151.0 ± 117.6 | Self-face vs. Self-house | **55.9**[a] | 1.5 | $10^{-4}$ |
| | | | Other-face vs. Other-house | 47.7 | 2.0 | 0.899 |
| | | | Self-face vs. Other-face | 50.2 | 1.8 | 0.415 |
| | | | Self-house vs. Other-house | 48.0 | 1.2 | 0.912 |
| | PPA | 1202.0 ± 60.1 | Self-face vs. Self-house | **55.4**[a] | 2.1 | $3.3*10^{-4}$ |
| | | | Other-face vs. Other-house | 49.5 | 1.7 | 0.601 |
| | | | Self-face vs. Other-face | 50.9 | 1.9 | 0.285 |
| | | | Self-house vs. Other-house | 48.5 | 2.5 | 0.810 |
| | FEF | 1967.0 ± 212.2 | Self-face vs. Self-house | 52.0 | 2.0 | 0.106 |
| | | | Other-face vs. Other-house | 50.0 | 1.6 | 0.495 |
| | | | Self-face vs. Other-face | 52.2 | 2.1 | 0.119 |
| | | | Self-house vs. Other-house | 51.6 | 1.8 | 0.136 |
| | SPL | 4327.6 ± 555.3 | Self-face vs. Self-house | 51.6 | 1.8 | 0.182 |
| | | | Other-face vs. Other-house | 52.5 | 2.2 | 0.074 |
| | | | Self-face vs. Other-face | 52.0 | 2.5 | 0.126 |
| | | | Self-house vs. Other-house | 49.1 | 1.5 | 0.700 |

Values in bold with [a] indicate significance after Bonferroni corrections for multiple comparisons

Thus, we compared four fixation patterns for each subject: Self-face, Self-house, Other-face and Other-house. The behavioral accuracy of target dot detection was 78.4% (SE 3.2%), with an accuracy of 82.1% (SE 3.7%) in detecting the target dot in the red central fixation, and an accuracy of 74.6% (SE 5.7%) in detecting the target dot during the gaze patterns. The mean latency was 190.8 ms (SE 15.0 ms) with a mean overall Spearman cross-correlation of rho(spear) = 0.272 (p < 0.001 for each subject, Spearman correlation testing). The lower cross-correlation value in Experiment 3, compared with Experiments 1 and 2, is at least partly due to the low numbers of saccades occurring in a short time window (0.6 s relative to 3 s in Experiments 1 and 2) and the higher speed of the gaze sequences (Supplementary Table 1).

Again, we conducted a multivariate classification analysis on the gaze patterns (x and y coordinates). As in the previous

**Table 3 Multivariate decoding results of early visual areas.**

| Experiment | ROI | N voxels | Classification | Mean (%) | SE (%) | p |
|---|---|---|---|---|---|---|
| Exp. 1 | V1 | 811 | Face vs. House | 50.9 | 1.2 | 0.228 |
| | | | House vs. inverted Face | 47.7 | 1.2 | 0.991 |
| | | | Face vs. inverted Face | 50.5 | 1.2 | 0.252 |
| | V2 | 685 | Face vs. House | 48.8 | 1.0 | 0.850 |
| | | | House vs. inverted Face | 49.1 | 1.1 | 0.832 |
| | | | Face vs. inverted Face | 50.5 | 1.0 | 0.310 |
| | V3 | 544 | Face vs. House | 49.3 | 0.9 | 0.665 |
| | | | House vs. inverted Face | 48.2 | 1.1 | 0.955 |
| | | | Face vs. inverted Face | 50.4 | 1.4 | 0.321 |
| | V4 | 516 | Face vs. House | 48.4 | 1.3 | 0.954 |
| | | | House vs. inverted Face | 49.4 | 1.0 | 0.722 |
| | | | Face vs. inverted Face | 48.7 | 1.3 | 0.894 |
| Exp. 2 | V1 | 811 | Face vs. House | 50.8 | 1.2 | 0.164 |
| | | | House vs. inverted Face | 48.6 | 1.4 | 0.884 |
| | | | Face vs. inverted Face | 47.9 | 1.9 | 0.962 |
| | V2 | 685 | Face vs. House | 52.0 | 1.5 | 0.022 |
| | | | House vs. inverted Face | 48.4 | 1.1 | 0.942 |
| | | | Face vs. inverted Face | 48.5 | 1.9 | 0.910 |
| | V3 | 544 | Face vs. House | 50.4 | 1.4 | 0.316 |
| | | | House vs. inverted Face | 48.7 | 1.2 | 0.890 |
| | | | Face vs. inverted Face | 48.3 | 1.7 | 0.935 |
| | V4 | 516 | Face vs. House | 51.7 | 1.4 | 0.035 |
| | | | House vs. inverted Face | 50.5 | 1.0 | 0.330 |
| | | | Face vs. inverted Face | 48.6 | 1.4 | 0.891 |
| Exp. 3 | V1 | 811 | Self-face vs. Self-house | 50.6 | 1.7 | 0.360 |
| | | | Other-face vs. Other-house | 51.3 | 1.8 | 0.223 |
| | | | Self-face vs. Other-face | 52.0 | 2.2 | 0.113 |
| | | | Self-house vs. Other-house | 50.9 | 2.0 | 0.271 |
| | V2 | 685 | Self-face vs. Self-house | 49.7 | 1.7 | 0.625 |
| | | | Other-face vs. Other-house | 53.2 | 1.4 | 0.036 |
| | | | Self-face vs. Other-face | 50.6 | 2.5 | 0.420 |
| | | | Self-house vs. Other-house | 51.4 | 2.1 | 0.182 |
| | V3 | 544 | Self-face vs. Self-house | 49.1 | 1.9 | 0.667 |
| | | | Other-face vs. Other-house | 53.1 | 1.8 | 0.038 |
| | | | Self-face vs. Other-face | 51.6 | 1.8 | 0.198 |
| | | | Self-house vs. Other-house | 51.6 | 2.0 | 0.155 |
| | V4 | 516 | Self-face vs. Self-house | 53.1 | 2.4 | 0.023 |
| | | | Other-face vs. Other-house | 52.9 | 1.7 | 0.046 |
| | | | Self-face vs. Other-face | 52.1 | 1.9 | 0.076 |
| | | | Self-house vs. Other-house | 52.1 | 1.2 | 0.096 |

experiments, the results showed a high discriminability of the gaze patterns of the Face versus House conditions across all subjects (64.4% mean accuracy, SE = 1.5%).

The repeated measures ANOVA with the factors ROI (FFA, PPA, FEF, SPL) and Fixation Pattern (Self-face, Self-house, Other-face, Other-house) in Experiment 3 yielded a significant main effect of ROI, $F(3,45) = 67.88$, $p < 0.001$, but no significant main effect of Fixation Pattern, $F < 1$. Although there was a significant interaction between ROI and Fixation Pattern, $F(9, 135) = 2.172$, $p = 0.028$, separate repeated measures ANOVA with only the factor Fixation Pattern did not reveal significant effects in any of the four ROIs, FFA: $F(3,45) = 1.22$, $p = 0.313$; PPA: $F(3,45) = 1.34$, $p = 0.274$; FEF: $F < 1$; SPL: $F < 1$. The univariate analysis on early visual areas did not show any differences among the different fixation patterns (Fig. 3c, see Supplementary Notes for the full ANOVA results).

The multivariate classification analyses showed that only self-generated face and house-specific gaze sequences could be discriminated, while the classification accuracy for Other face vs. house fixation sequences did not exceed the permutation-based chance level (Table 2 and Fig. 4c). This selective pattern—also observed at the single-subject level (Supplementary Fig. 1)—was only observed in FFA and PPA. Neither the eye-movement-defined frontoparietal ROIs nor the early visual (V1–V4) ROIs showed any significant above chance classification accuracy in any of the contrasts between self/other face/house patterns. Moreover, the classification accuracies for self-generated gaze tracks (Self-face vs. Self-house) were higher in FFA and PPA than V1–V4, $p = 0.002$ (permutation testing).

For both FFA and PPA, a post hoc permutation-based comparison showed that the classification accuracy of the Self-face versus Self-house comparison was higher than the classification accuracy of the Other-face versus Other-house comparison (FFA: $p = 8.5 \times 10^{-4}$, PPA: $p = 0.018$). The same contrast yielded no differences in classification accuracies in FEF and SPL or in the early visual areas (all $p > 0.210$, permutation testing). Moreover, the classification accuracies for self-generated gaze tracks were higher in FFA and PPA than in FEF and SPL ($p = 0.019$, permutation testing)

The classification analysis on the eye-movement data ($x$ and $y$ coordinates) while subjects were following the gaze patterns showed a mean accuracy of 57.0% (SE = 1.4%) for self-face (SF) vs. self-house (SH), and a mean accuracy of 54.5% (SE = 1.7%) for other-face (OF) vs. other-house (OH), a mean accuracy of 63.3% (SE = 2.7%) for SF vs. OF, and a mean accuracy of 60.3% (SE = 1.7%) for SH vs. OH. The eye-movement patterns in

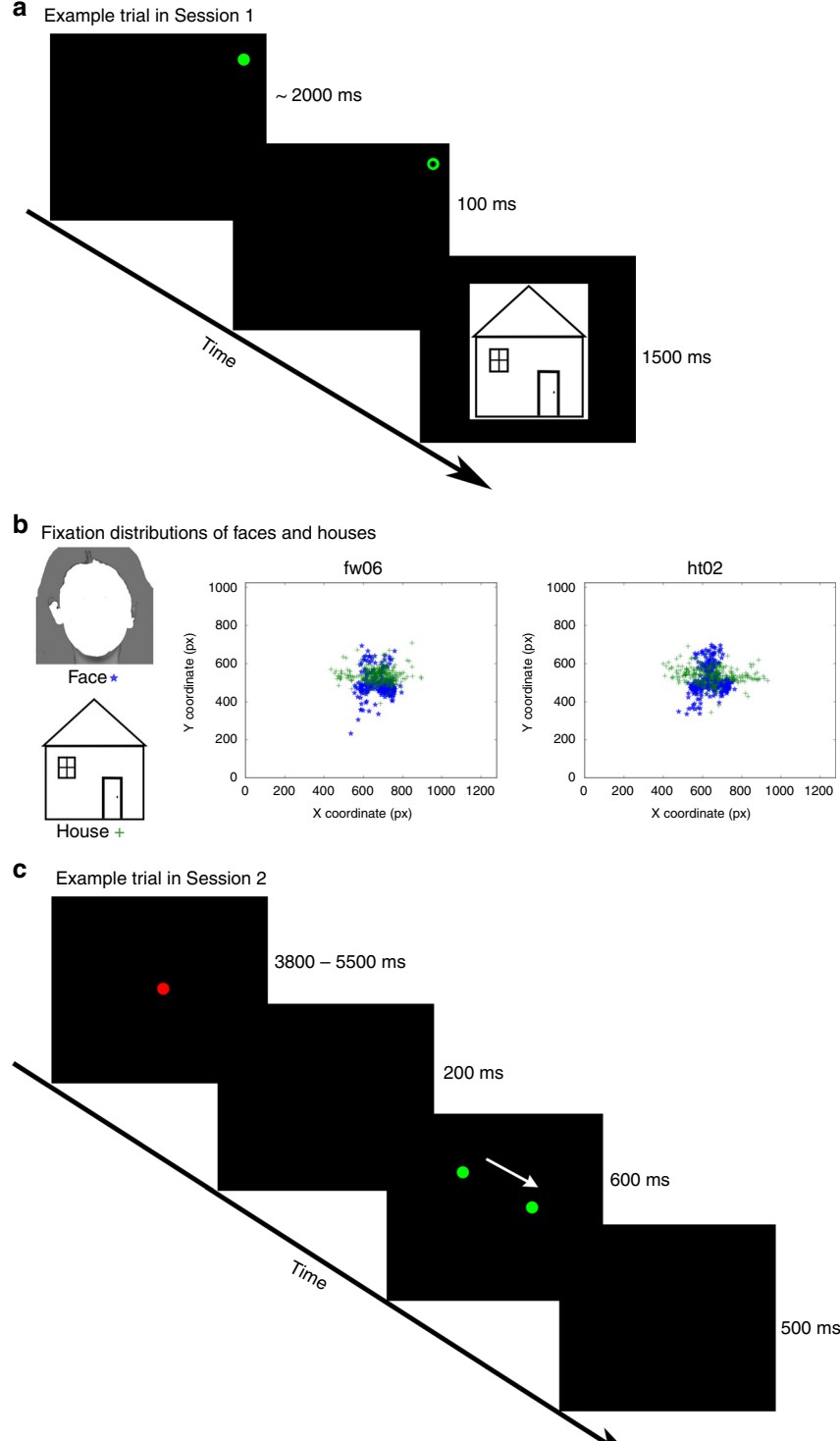

**Fig. 5 Experimental design and fixation patterns in Experiment 3. a** An example trial in the gaze-recording session of Experiment 3. In each trial, participants were asked to fixate the green dot which was located at one of the four screen corners, and detect a small black dot inside the green dot. When the face or house appeared, participants had to indicate if the current picture was new or one of the previously presented pictures. This task required attentive viewing for face or house recognition. **b** A face example (shown as contour here for privacy protection) and a cartoon example of the house from the picture set (left), and the fixation patterns collected from the 600 ms interval after stimulus onset from two participants (collapsed over all face and house pictures, right). These fixation patterns were used for the scanner session of Experiment 3. **c** An example trial in the scanner session of Experiment 3 that was conducted 1 week after the gaze-recording session. A red central fixation dot was followed by a sequence of green moving dots, representing the respective fixation pattern. In 20% of the trials, equally probable in the four conditions (Self-face, Self-house, Other-face, Other-house), a small black dot centered inside the fixation dot (equally probable for the red central fixation dot and the green moving dot) served as target. Subjects were instructed to respond with a forced choice button press to target presence. The white arrow only indicates the track of the green dot, but was not presented in the actual experiment.

distinguishing OF vs. OH, SF vs. OF, SH vs. OH dot patterns did not differ from the eye-movement patterns in distinguishing SF vs. SH dot patterns, $p = 0.157$ (permutation testing). Therefore, the FFA- and PPA-specific classification of SF vs. SH cannot be reduced to the physical properties of the stimulus-driven eye-movement data.

For the self-generated gaze tracks, we further analyzed sensitivity and bias to further characterize the nature of their representation. The FFA showed a higher sensitivity for Self-face vs. Self-house discrimination compared with parietal or early visual ROIs (Supplementary Figs. 2 and 3). The same was true for the PPA. However, PPA responses were biased towards face classification, with a high number of false alarms, whereas no bias was observed in the other areas (Supplementary Fig. 2). One caveat may be that Experiment 3 was optimized for face perception, thus perhaps leading to a disadvantage for the PPA, which showed no bias in Experiments 1 and 2.

We further investigated if participants' ability to follow the dots affected classification sensitivity (area under ROC curve, see methods and Supplementary Fig. 2). For the FFA, the individual classification sensitivity of the Self-face versus Self-house comparison showed a positive correlation with the individual cross-correlations between the gaze-track position and the actual eye position (Pearson $r = 0.627$, $p = 0.009$; Fig. 4d, left). Thus, those participants who most closely followed the dots with their fixations also showed the most distinct activation patterns in the FFA. This underlines that the activation patterns depend on eye movements, not just on passively viewing the stimulation sequences without following them.

We also asked if the individual sensitivity in discriminating self-face vs. self-house gaze tracks in FFA was affected by participants' distinct eye-movement patterns for face viewing. Specifically, we measured the consistency of the first fixation locations for face viewing within each subject, which has been reported to be unique and stable among individuals [23,24]. We calculated the variation of the first fixation locations acquired from each single participant when they were looking at faces. We found that this individual variation negatively correlated with the individual classification sensitivity in FFA (Pearson $r = 0.536$, $p = 0.033$; Fig. 4d, right). In other words, participants with more consistent first fixation location in perceiving faces showed more distinct FFA activation patterns.

Did the participants recognize their own gaze-tracks and could such an explicit memory have contributed to the better discrimination of the associated activation patterns? This possibility was not supported by the results from a follow-up experiment in which the gaze tracks were again played to participants and they were alerted to the fact that these were gaze tracks recorded from themselves and from other participants. In spite of this information, participants were unable to differentiate their own gaze-track from another observer's gaze track (48.5% mean accuracy, SE 2.5%; area under the ROC curve: mean 0.46, SE 0.03; supplementary Supplementary Fig. 4).

## Discussion
We have shown that category-specific gaze patterns are represented in high-level visual cortex. Retracing gaze paths originally recorded while viewing faces or houses led to differential activation patterns in FFA and PPA. When gaze patterns were followed over a period of 3 s, category-specific activation patterns were not only observed in FFA and PPA, but also in the more anterior visuomotor areas SPL and FEF. However, when gaze tracks were restricted to 600 ms, allowing on average two saccades, which has been found to suffice for face identification[28], differential activation patterns elicited by face or

house related gaze tracks were restricted to FFA and PPA. These activation patterns appear to represent individual visuomotor programs in that only the participants' own gaze tracks could be discriminated, but not the gaze tracks of other participants.

FFA and PPA have been shown in numerous studies to be activated strongly by viewing pictures of faces and houses, respectively, even when these stimuli are presented tachistoscopically, discouraging the use of eye-movements (as in our localizer tasks). Nevertheless, looking at a face or a house usually goes along with distinct gaze paths. Here, we investigated whether FFA and PPA-activations might in part be due to differences in these gaze patterns, even when they occur in the absence of face or house presentations. Participants were following a small circle on an otherwise empty screen with their gaze, thereby creating a gaze sequence typical for looking at a face or house (or inverted face), unaware of this. This lack of knowledge, together with an attention-demanding task—detecting a small dot inside the target circle—makes it unlikely that the activation was elicited by imagery of faces or houses.

Even the more unspecific effects of face and house-related gaze tracks are noteworthy in that they show how the motor programs carried out during looking at a face or house can contribute to activation differences in the FFA and PPA. The more widespread dispersion of fixation locations during house viewing led to an increased univariate BOLD response in these areas. For studies of the PPA, this may serve as a caveat that eye movements need to be controlled so that activation elicited by eye-movements are not confounded with other aspects of house or scene viewing. The higher dispersion of house-related gaze patterns appears to contribute to the stronger activation of the PPA when viewing houses than faces. There are parallels between this finding and previous reports of retinotopic differences between FFA and PPA[29]. The stronger representation of the periphery in the PPA might be linked to a higher propensity to elicit saccades into the periphery by PPA-neurons.

When the dispersion of fixation locations was equalized for face and house gaze tracks, univariate activation differences were no longer observed but face, inverted face and house-related gaze tracks could still be discriminated based on multivariate analysis of activation patterns. This discrimination, however, was not only possible in FFA and PPA, but also in SPL and FEF, areas involved in eye movement control. This was not unexpected, because if FFA and PPA contain integrated object-action representations, they are likely to interact with more anterior visuomotor areas. In contrast, gaze-track category did not lead to distinct activation patterns in early visual areas, suggesting that actions are bound to object representations rather than to low-level visual features. However, on the basis of Experiments 1 and 2 we could not completely rule out that face and house-related gaze tracks still differed in some basic features, e.g. the more orderly localization of fixations in the triangle between eyes and mouth during face viewing compared with the more scattered fixations during house viewing. Related to this, different gaze tracks might have been confounded with complex visual stimulation differences that may not have been visible in V1–V4, but may have contributed to the activation differences in the FFA and PPA as well as SPL and FEF.

In order to eliminate these potential confounds as far as possible in Experiment 3, we first reduced the fixation locations presented to our participants to the minimum needed for face recognition[28]. This led to a reduction of stimulation from 3000 ms in Experiments 1 and 2 to only 600 ms in Experiment 3, thereby eliminating potential noise that is unrelated to face recognition. In addition, we used the knowledge about inter-individual differences in face viewing to create a design that allowed us to compare identical gaze tracks that only differed in

that these tracks were self-generated for one subject but not self-generated for the next subject. When we compared face and house related gaze tracks in this way, eliminating all stimulation or gaze parameter differences between self and other generated gaze tracks across participants, the discrimination of face and house-related gaze-tracks was only possible in the FFA and PPA and only for self-generated gaze-tracks. Classification accuracy increased with the accuracy with which participants' gaze followed the dots, demonstrating that the classification is based on eye movements, not changing stimulus locations. Nevertheless, it might be worthwhile to further investigate the underlying factors contributing to accurate and poor gaze following. Blinks were eliminated as causes for poor gaze following, however, the quality of gaze following may still reflect fluctuations of vigilance. Therefore, a replication with fixed central fixation may be worthwhile. Ideally, a replication of our dot-following experiments with the instruction to keep central fixation should eliminate any eye-movements. In this hypothetical experiment, we may not expect a representation of eye movement patterns. However, even with central fixation, microsaccades may occur that follow the dot sequence. The question if such microsaccades may be represented in high-level visual cortex may be interesting in itself.

In agreement with the importance of the first fixation location for individual differences in face perception[22,23], we found that the classification sensitivity of the FFA was highest for those subjects with the most consistent first fixation location for face tracks. These findings cannot be explained by conscious knowledge or even hunches about the nature of the gaze tracks, as participants were not aware of the nature of the dot movements in the post-experimental interview and they could not discriminate self vs. other tracks when told to do so in a forced choice post-test.

Thus, we have evidence of individual representations of category-specific eye movement patterns in the FFA and PPA. Our results differ from previous reports of neuronal modulation by gaze parameters in early visual cortex in that not basic parameters like gaze direction[19] were represented, but complex sequences of eye movements. We could not discriminate category-specific gaze tracks in early retinotopic areas in any of the three experiments. Even in Experiment 1 with its clear differences in the distribution of fixation locations between faces and houses and concomitant overall activation differences was the classification of category-specific gaze track activation patterns only possible in high-level visual cortex (FFA and PPA) and the visuomotor areas (SPL and FEF). Finally, Experiment 3 showed that stimulation differences could not account for the representation of self-generated gaze patterns in the FFA and PPA. Thus, it appears that FFA and PPA contain more selective information about individual face-scanning patterns than the early visual cortices that represent aspects of eye movements that are shared across individuals.

The classification of self-generated gaze tracks in the FFA and PPA relied on increased sensitivity for face-tracks. While this was expected for the FFA, the PPA also showed high sensitivity for faces. However, the PPA classification results were biased towards face-tracks, leading to a high number of false alarms, whereas classification of FFA activation patterns was unbiased. A possible reason for the bias observed in the PPA may be that Experiment 3 was optimized for face viewing, particular in its restriction to the first 600 ms. Thus, the current results may underestimate the specificity of the PPA response to optimal house viewing gaze tracks.

On first sight, it might seem puzzling that other participant's face- versus house-related gaze tracks could be discriminated in Experiments 1 and 2, but not in Experiment 3. However, as noted, gaze tracks in Experiment 3 were considerably shorter (a fifth of the duration) in Experiment 3. On the one hand, this reduction

reduced the gaze-specific signal that could be picked up, so that only optimal—self-generated—gaze tracks could be discriminated. On the other hand, the discrimination of other observers' gaze tracks in the first two experiments might indicate that there is information in the gaze tracks beyond the first 600 ms that can be used to discriminate face and house-specific tracks. Characterizing these differences is clearly a goal for further research, particularly, as we cannot rule out that subtle visual stimulation differences may have contributed to the classification in Experiments 1 and 2.

While we consistently found that face and house-specific gaze tracks are represented in FFA and PPA, the function of this representation remains to be investigated further. Our experiments were quite artificial in that participants followed a dot movement that replayed a fixation sequence originally shown during face or house viewing. This, of course, is not what happens during actual viewing of an object or a scene, where the eye movement pattern is influenced by external as well as internal factors[30]. This raises the question if our experiments activated a motor program represented in FFA and PPA that helps guide eye movements during actual face or house viewing or if these areas analyze eye movement patterns—generated by parietal and frontal neurons—to help distinguishing faces from houses in addition to the many other visual cues that are normally available when we look at a face or house (but which were missing in our experiments). Our results may rather favor the latter alternative because our participants had no way to internally generate a face or house-related eye movement pattern but relied completely on external stimuli to guide their gaze. Nevertheless, we cannot rule out that FFA and PPA may also be involved in generating category-specific eye movement patterns. Future patient studies might be instructive on this issue—do FFA lesions lead to disturbed gaze-patterns during face viewing or is face recognition disturbed in these patients although gaze patterns are normal? Existing case studies of gaze patterns in patients with acquired prosopagnosia are unfortunately not specific enough to answer this question, showing overall normal (but see ref. [31]), but for familiar faces less predictable, eye movement patterns[32–34]. Interestingly, a patient with acquired prosopagnosia due to a large ventral occipitotemporal lesion showed aberrant gaze patterns during (non-facial) object naming and object recognition tasks[35] that may support the idea of an active contribution of occipito-temporal cortex to object-specific gaze patterns

Another open question is why self-generated face and house-associated gaze tracks led to distinct activation patterns in the FFA and PPA, but not in SPL and FEF, as might be expected if high-level sensory and motor areas share a common code linking perception and action[5]. One potential explanation might be that individual eye-movement programs in SPL and FEF might only be activated in active vision—as during normal face viewing—but not while passively following a dot, as in our experiments. The functional role of gaze pattern representation in high-level visual cortex and its interaction with traditional visuomotor areas thus remain to be investigated.

## Methods

**Subjects**. In Experiment 1, 21 subjects (11 of them female) participated. Their average age was 24.6 (SD = 2.4) years. Experiment 2 included 18 subjects (7 of them female) with an average age of 27.2 (SD = 5.0) years. Experiment 3 included 19 subjects (7 females) with an average age of 26.6 (SD = 4.1) years. Data from one subject was excluded due to excessive head movements during scanning (>3 mm). Data from another two subjects were excluded from further analysis due to poor recording of eye-tracking data in the scanner, leaving 16 participants (6 females) with an average age of 27.2 years (SD = 4.1). For one of the remaining 16 participants, data from one run was excluded due to excessive head movements during the scanning of this run (>3 mm). All subjects reported normal or corrected to normal vision.

All experiments followed the principles of the Declaration of Helsinki. Informed written consent was obtained prior to experiments in accordance with the protocols approved by the local Ethics Committee of the Medical Faculty of Otto-von-Guericke-University, Magdeburg. Subjects in Experiments 1 and 2 received €18, and Subjects in Experiment 3 received € 40 as compensation.

**Design and procedure.** In Experiments 1 and 2, we recorded gaze patterns of two naive subjects while they were freely looking at 42 unique gray-scaled images of faces and houses (10 × 10 of visual angle) that were presented for 3000 ms duration (Supplementary Table 1). These subjects did not participate in the main study. Their gaze patterns were only used as stimuli in the main experiment. Images were presented centrally on a light gray background. Eye positions were acquired using a video-based EyeLink 1000 eye tracker (SR-Research, Canada) with a temporal resolution of 1000 Hz. After removal of eye blinks using the python-based module cili (https://github.com/beOn/cili), fixations longer than 200 ms were identified. Fixations were defined as interval in which the velocity of the eye movements did not exceed 30° of visual angle per second, as 30°–40° per second were suggested to identify a saccade[36].

Each subject of the main experiment completed a training session to become familiar with the task. During the training session, the subjects were asked to follow a dot on the screen with their gaze. This dot started at the central fixation position and followed the reconstructed fixation sequences of the freely observing subjects. For each category (faces and houses), 56 out of the 84 recorded fixation patterns were randomly selected and presented during the training. As an additional control condition, the face fixation patterns were flipped vertically along the horizontal axis, thereby serving as a pseudo-inverted face condition. In 50% of the trials, with equal probability in all three conditions, a small gray dot centered inside the black part of the fixation dot served as target. The subjects were instructed to always direct their gaze to the fixation point and to respond by pressing the left mouse button at the end of each trial whenever they perceived the target, and the right button whenever they did not detect a target. Because of the small size and luminance change, fixation of the dot was necessary to detect the target. To prevent subjects from interrupting the fixation sequence task after potential detection of a target in an early stage of the trial, the target was presented in the last fixation position of each trial that exceeded the fixation duration of 300 ms (see Fig. 1b).

For Experiment 2, the fixation patterns were altered such that the mean value and the standard deviation of x- and y-coordinates of the house viewing patterns were matched to the values of the face condition, thereby creating an even dispersion of the fixation patterns across categories. Firstly, for both face and house fixations, the mean ($M_f$, $M_h$) and standard deviation ($SD_f$, $SD_h$) of the Euclidean distances between the fixations and the center of the screen were obtained. Then, for house fixations, the x,y coordinates of each fixation were transformed such that the mean ($M_h'$) and standard deviation ($SD_h'$) of the Euclidean distances matched the face fixations, with the following equations:

$$(x_{hi}, y_{hi}) = ((x_{hi}, y_{hi}) - M_h)/SD_h, \qquad (1)$$

$$(x'_{hi}, y'_{hi}) = (x_{hi}, y_{hi}) * SD_f + M_f, \qquad (2)$$

where $x_{hi}$, $y_{hi}$ indicate the x, y coordinates of the ith house fixation before the transformation, and $x_{hi}'$, $y_{hi}'$ indicate the x, y coordinates of the ith house fixation after the transformation. This procedure was performed separately for each of the two subjects. Additionally, we replaced the inverted face fixation sequences of Experiment 1 by fixation sequences recorded while new subjects looked at real inverted faces (Supplementary Table 1). Furthermore, we slightly increased the temporal resolution of the fixation patterns by lowering the minimal fixation duration to 150 ms, so that the gaze sequences resembled the real eye movements more closely.

In the scanner session, the experimental design and task was similar to the training session. The same fixation patterns were presented as in the training. We increased the luminance contrast of the target dot in comparison to the training session to compensate for the aggravated condition of performing psychophysical tasks in the scanner and to avoid discouraging the subjects. The probability of target occurrence was lowered to 25%. The presentation of the 168 fixation patterns was balanced to achieve an equal number of presentations for each gaze pattern category. Gaze patterns of each category were presented 56 times during the course of the experiment. The conditions were accordingly balanced and the presentation sequences were individually randomized. The mean inter-stimulus-interval was 6.2 s. The duration of the experiment was 26 min.

The main experiment was followed by a standard localizer for FFA and PPA. The same gray scaled images of houses and faces that we used to derive the fixation sequences were presented. Faces were shown both upright and inverted. Subjects were instructed to look passively at the pictures while keeping central fixation. Faces, inverted faces and houses were presented in seven blocks each. Each block lasted for 15 s. Pictures were flashed for 200 ms to discourage eye movements, followed by a blank screen for 800 ms. Each block was followed by a 12 s baseline in which no object was presented. The localizer run started with a 12 s fixation baseline.

In Experiment 3, we recorded gaze patterns of all subjects while they were looking at faces and houses in an N-back task (Fig. 5a) that took place 1 week before the fMRI scanning. In this task, each trial began with a green dot (0.2° of

visual angle in diameter) at one of the four corners (15° from the center of the screen) of a black screen. The duration of this green dot was randomly selected from a normal distribution (mean = 2 s, SD = 0.1 s). In order to force participants to fixate the dots, in 20% of the trials, a small black dot (0.05° in diameter) was presented together with the green dot for 0.1 s, with the black dot localized at the center of the green dot. Subjects were asked to detect the black dot by pressing the 'y' button in the keyboard using the left index finger. Looking at the peripheral dots served the purpose to enable us to record the first fixation on the subsequent face or house pictures without biasing this fixation position by central fixation. After the disappearance of the green dot, a picture of a house or a face was presented at the center of the screen, and remained on the screen for 1.5 s. The height of the picture was fixed at 18° of visual angle, to ensure that the eye-to-mouth distance in the face picture was 6°. This is about the size of a face in a conversational situation[37]. All subjects went through 14 blocks, with a picture set of seven houses and seven faces repeatedly presented in each block (one picture per trial). They were required to memorize the pictures in the first block, and detect if a new picture was presented in the following 13 blocks (one or two occurred in each block) by pressing the 'm' button in the keyboard using the right index finger.

For the eye movement data, eye blinks were first removed. Trials without any valid fixation events, and trials with fixation localized beyond the region of the picture were also excluded. In each of the remaining trials, fixation events were resampled in a time window of 0–0.6 s post-picture onset. During this 0.6 s time window, a fixation was identified as a gaze event if its duration was longer than or equal to 0.1 s, while identified as a non-gaze event if its duration was shorter than 0.1 s. This non-gaze event was represented by a blank screen in the scanner session, simulating the break between fixations to make the gaze tracks more naturally. After that, trials with less than two gazes were excluded. The remaining trials were used in the scanner session. The gaze coordinates were proportionally transformed corresponding to the screen resolution in the scanner such that the sizes were matched for the lab session and the scanner session.

In the scanner session, each participant was asked to follow dots represented his/her own gaze patterns as well as dots represented another observer's gaze patterns in independent trials, rendering four experimental conditions: Self-face, Self-house, Other-face, and Other-house. All stimuli were presented through an LCD projector onto a rear screen located behind the subjects' head. The participant viewed the screen (screen resolution: 1280 × 1024) via an angled mirror mounted on the head-coil of the MRI setup. Each trial began with a red dot on a black screen, which remained at the center of the screen for 3.8–5.5 s. During this time interval, a black dot (0.05° in diameter) appeared at a random time point and lasted for 0.1 s. This black dot was presented at the center of the red dot, and occurred in 10% of all trials. As a fixation check, participants were asked to detect the black dot by pressing the button using the left index finger. After a blank screen of 0.2 s, the gaze track, which was represented by a sequence of green dots, was presented on the screen. According to our experimental design, the gaze track in each trial lasted for 0.6 s, during which 2–5 green fixation dots (0.2° in diameter) were presented sequentially. For a gaze event, a green fixation dot was presented on the screen. For a non-gaze event, a blank screen was presented. The durations of the gaze event and the non-gaze event were the same as the durations in the gaze-recording session. The coordinates of the gazes (i.e., green dots) were proportional to the coordinates in the gaze-recording session such that the gaze tracks were presented with the same visual angles. In 10% of the trials, a black dot (0.05° in diameter) appeared at the center of one green dot and lasted for additional 0.1 s. Participants were asked to detect the black dot by pressing the button using the left index finger. The black-dot event occurred with equal probability for the four experimental conditions (Self-face, Self-house, Other-face, Other-house). After the offset of the gaze track, a blank screen of 0.5 s was presented to separate the gaze track from the red central dot in the next trial (Fig. 5c). There were 14 scanning runs, with 28 trials (seven trials for each condition) in each run. Trials from different conditions were mixed and presented in a random order. To identify FFA and PPA, participants completed 4 scanning runs of a visual localizer task[38], during which pictures of different object categories (faces, houses, objects, human bodies without head, outdoor scenes and scrambled pictures) were presented[39].

After the fMRI scanning, participants were asked if they could recognize the pattern of the moving dots during the main experiment. None of them reported the recognition of the moving dots or the connection between the two sessions. To further test the awareness of the gaze-tracks, we conducted a follow-up experiment outside the scanner which had the same design as the experiment in the scanner except that the subjects were told that the dots represented gaze tracks and asked to make a 2-alternative-forced choice response in each trial to judge whether the moving dots represented their own gaze-track or another person's gaze-track. Only eight subjects participated in this follow-up experiment because other subjects either had moved or lost touch with us.

**MR-parameters.** In Experiments 1 and 2, MR-data were acquired at a 3T Siemens Trio MR-scanner equipped with an 8-channel head-coil. For anatomical coregistration, a T1 weighted image was recorded before the functional imaging (192 sagittal slices, 256 × 256 1 mm isotropic voxels, TR = 2500 ms, TE = 4.77, FA = 7°). The same echo-planar imaging sequence was used for the main experiment as well as for the localizer (34 transversal slices, 3.5 mm isotropic voxels, matrix size = 64 voxels, TR = 2000 ms, TE = 30 ms, flip angle = 80°,

interleaved slice acquisition). During the main experiment, 780 volumes were recorded, while the localizer lasted for 330 volumes.

In Experiment 3, MR-data were acquired at a 3 T Philips Achieva dStream MR-scanner equipped with a 32-channel head-coil. For anatomical coregistration, a T1 weighted image was recorded before the functional imaging (192 sagittal slices, $256 \times 256$ 1 mm isotropic voxels, TR = 2300 ms, TE = 4.65 ms, FA = 8°). The same echo-planar imaging sequence was used for the main experiment as well as for the localizer (35 transversal slices, 3 mm isotropic voxels, matrix size = 80 voxels, TR = 2000 ms, TE = 30 ms, flip angle = 90°, ascending slice acquisition). During the main experiment, 14 runs of 97 volumes were recorded, while the localizer lasted for 4 runs of 156 volumes.

**Evaluation of the eye-tracking dataset**. To evaluate the ability of the subjects to perform the task, in Experiments 1 and 2, the correspondence between the actual eye movement data of the training session and the course of the fixation point was estimated. After removal of eyeblinks, the dataset was separated in trials starting from 200 to 3700 ms after onset of the fixation sequence. Per subject, we applied a cross-correlation method that maximized the correlation between the eye movement data and the actual fixation point position by a temporal shift. Within a window of 400 ms after stimulus onset the latency that maximizes the correlation between fixation dot position and eye movement data was determined in each trial separately for the $x$- and $y$-axis. As general evaluation score, we computed Spearman's correlation between the model, which was shifted by the mean latency of the maximal correlation, and eye movement data over the entire dataset (Fig. 1c). The same cross-correlation was conducted in Experiment 3 except that the dataset was separated in trials starting from the onset of the central fixation and the offset of the fixation pattern (moving dot).

The differences in gaze patterns between the categories were evaluated by means of a multivariate classification analysis. $X$ and $y$ coordinates in the interval from 200 ms to 3700 ms after stimulus onset of each trial were used as features. Missing data due to eyeblinks were substituted by a 1° (linear) spline interpolation. Trials of both subjects were pooled together. The samples of each condition were equally distributed across six chunks. Accordingly, the dataset was structured as follows: instances (2 subjects × 6 chunks × 3 classes × 7 samples) × features (3500 $x$-coordinates + 3500 $y$-coordinates). Before classification each feature was chunk-wise z-scored. Based on this dataset we calculated a leave-one-out cross-validated classification using a support vector machine classifier. In order to prevent overfitting of the classification model we reduced the number of features by means of feature selection, only including the features with the 10% highest $F$-values between classes in the training dataset. Mean accuracies (percentage of correct assignments of samples of the test dataset to the corresponding class) over all cross-validating steps are reported. Similar multivariate classification analysis was conducted on the $x$ and $y$ coordinates in the interval of the gaze tracks in Experiment 3. Given that the gaze patterns in Experiment 3 were presented individually, this classification analysis was conducted for each individual subject, and the mean decoding accuracy with SE across all subjects were reported.

In Experiment 3, the variation of the first fixation locations while subjects were looking at faces (i.e., the location of the first dot in the self-face condition in the scanner session) was also calculated, as previous studies have documented unique and stable individual first fixation locations for faces[22–24]. Here the variation was obtained by averaging the Euclidean distances among the first fixation locations across all trials for each subject. A shorter mean Euclidean distance indicates a more consistent first fixation location for face viewing. Correlation analysis was conducted between this variation and the sensitivity in discriminating self-face vs. self-house in FFA to investigate if the discrimination in FFA was linked to how the observer explored faces.

**Localizer**. To identify FFA and PPA-ROIs, individual brains were masked using the Temporal Fusiform Cortex, posterior division, and Temporal Occipital Fusiform Cortex areas of the Harvard-Oxford Cortical Structural Atlas. All voxels inside this area were considered in the ROI-Analysis.

In Experiments 1 and 2, voxels were labeled face-sensitive if they reached positive $z$-values in the localizer contrast 'Face > House' and house-sensitive if they reached negative $z$-values (Fig. 2a, b). SPL and FEF were identified on the base of activation elicited by eye movements in the main experiments. SPL was defined as voxels with positive $t$-scores (i.e., a threshold of $t > 0$) for the contrast 'Face + House + inverted Face' inside of the superior parietal lobule (defined by the Harvard-Oxford Cortical Structural Atlas). FEF was defined by the same contrast by cuboids spanned between the MNI-coordinates $x + -52$, $y$ 0, $z$ 68, and $x + -18$, $y -18$, $z$ 44 in the region around the junction of the precentral and superior frontal sulci[40]. These areas surround FEF bilaterally according to the meta analysis (381 studies) on http://neurosynth.org with the feature name 'eye'[25].

In Experiment 3, voxels were labeled as FFA in the localizer contrast 'Face > all other pictures (houses, scenes, bodies, objects, scrambled pictures)' at a threshold of $z > 1.64$, and labeled as PPA in the localizer contrast 'House + Scene > all other pictures (faces, bodies, objects, scrambled pictures)' at a threshold of $z > 1.64$ (22; Fig. 2c). SPL and FEF were defined in the same respective regions as Experiments 1 and 2, based on the activated voxels in the contrast 'Self-face + Self-house + Other-face + Other-house' at a threshold of $z > 1.64$.

In all of the three experiments, early visual areas (V1–V4) were defined by the probability maps in the Juelich Histological Atlas[27]. To avoid overlap between these areas, voxels in these maps were thresholded based on the probability value in the way that only voxels with high probability were included for the further ROI analysis (Fig. 2d). Specifically, voxels in V1 and V2 were thresholded by 90%, i.e., voxels with the probability below the 90% percentile were excluded; and voxels in V3 and V4 were thresholded by 80%. In all of the three experiments, the value of each voxel in all ROIs was reset to 1 such that the different patterns of results in different ROIs cannot be reduced to the different initial values of the voxels. All of the above-mentioned ROIs were bilateral.

**Univariate analysis**. For the univariate analysis, we used FEAT (FMRI Expert Analysis Tool) Version 5.98, part of FSL (FMRIB's Software Library, www.fmrib. ox.ac.uk/fsl). Time-series were statistically analyzed by using FILM with local autocorrelation correction[41]. Pre-processing consisted of motion correction using MCFLIRT[42], brain extraction using BET[43], spatial smoothing using a Gaussian kernel of FWHM 5 mm, grand-mean intensity normalization of the entire 4D dataset by a single multiplicative factor, and highpass temporal filtering (Gaussian-weighted least-squares straight line fitting, with sigma = 50 s).In order to co-register the high-resolution structural T1 image and the functional images, a linear transformation with six degrees of freedom allowing translation and rotation was applied (FLIRT[44]). The 12 df linear transformation from high resolution structural to MNI-standard space was then further refined using FNIRT nonlinear registration[45,46].

**General linear model**. In Experiments 1 and 2, three regressors were defined for the conditions 'Face', 'inverted Face' and 'House' by the onsets of the stimuli with the individual length of the saccade sequence duration convolved with canonical double gamma HRF. To deal with activity caused by response button presses, separate regressors for 'hits', 'false alarms', 'correct rejections' and 'misses' were added to the model. The first derivative of each regressor was introduced as regressor of no-interest to reduce error variance due to inter-voxel differences of temporal dynamics of the HRF. In order to reduce the influence of head motion on signal changes, we added the 6 parameters of motion correction to the GLM. The stimulation of the localizer was modeled as a block design. Standardized parameter estimates were extracted using FeatQuery. Differences in parameter estimates were tested by an ANOVA with the within subject factors ROI (FFA, PPA, FEF, SPL) and Fixation Pattern (Face, inverted Face, House).

The construction of GLM in Experiment 3 was similar to that in Experiments 1 and 2. Four regressors were defined for the conditions 'Self-face', 'Self-house', 'Other-face', and 'Other-house' by the onsets of the gaze track with the length of the gaze-track sequence duration (0. 6 s). Differences in parameter estimates were tested by an ANOVA with the within subject factors ROI (FFA, PPA, FEF, SPL) and Fixation Pattern (Self-face, Self-house, Other-face, Other-house), and an ANOVA with the within subject factors ROI (V1, V2, V3, V4) and Fixation Pattern (Self-face, Self-house, Other-face, Other-house).

**Multivariate pattern analysis**. In order to detect differences of spatial-temporal signal patterns elicited by class-specific gaze patterns we performed a multivariate pattern classification. The analysis was realized using the Multivariate Pattern Analysis in Python toolbox (PyMVPA[47]). The data of each subject and ROI were separately analyzed. fMRI-data were motion corrected, and smoothed with a 4 mm FWHM Gaussian kernel using the Nilearn Python package to reduce noise while minimizing impact on fine-scale signal patterns[48].

In Experiments 1 and 2, each gaze pattern event was represented by the data samples acquired in the interval of 2–10 s (4 volumes) after event onset to take the temporal dynamics of the BOLD signal into account. The data samples in each ROI were divided into two halves. The data sample in each half was voxel-wise Z-scored across all events, and each event-related data sample in each half was then estimated into one β score with the HRF model. Accordingly, the dataset was structured as: instances (3 classes × 2 halves × 28 β scores) × features (voxel number within ROI). The same procedure was conducted in Experiment 3 except that the sample duration of the gaze-tracks was reduced to ~600 ms in Experiment 3 to focus on the initial fixations that are vital for face recognition. To accommodate the corresponding BOLD signal to this short interval, the seven data samples of each event in each run were estimated into one β score with the HRF model, rendering a structure of the dataset as: instances (4 classes × 14 runs × 1 β score) × features (voxel number within ROI). For all the three experiments, we calculated a cross-validated classification using a linear support vector machine as classifier, with the order of the two halves used as training and test dataset counterbalanced. We used half-partitioning here instead of leave-one-run-out for cross-validation to avoid that the test dataset is smaller than the training dataset, and to maintain comparable temporal correlation of the data within both the training dataset and the test dataset[49].

Mean accuracies (percentage of correct assignments of samples of the test dataset to the corresponding class) over all cross-validating steps are reported. The decoding was separately conducted for the pair-wise classifications. In Experiments 1 and 2, the pair-wise classifications were: Face versus House, Face versus inverted Face and House versus inverted Face. In Experiment 3, the pair-wise classifications

were: Self-face versus Self-house, Other-face versus Other-house, Self-face versus Other-face, and Self-house versus Other-house. On the individual level, chance distribution of the decoding accuracies was acquired separately for each of the four pair-wise classifications by performing 100 permuted classifications with a Monte Carlo technique[50]. For each participant, one accuracy was randomly selected from the set of accuracies obtained with a permuted model, and these individual chance accuracies were averaged into a group chance accuracy. This procedure was repeated $10^5$ times with replacement of the individual accuracy, resulting in a distribution of $10^5$ permuted group accuracies. Significance testing was conducted by calculating the probability of the unpermuted mean decoding accuracy across participants in the distribution of the permuted group accuracies (one-tailed). Multiple testings for different pair-wise classifications in each ROI were corrected with Bonferroni method.

The sensitivity in terms of the area under the ROC curve (AUC), and the decision bias[51] of the ROIs in discriminating the gaze-tracks (Face versus House in Experiments 1 and 2, Self-face versus Self-house in Experiment 3) were also calculated. The AUC was calculated using the formula $AUC = \Phi((\Phi^{-1}(H) - \Phi^{-1}(F))/\sqrt{2})$, where $H$ indicates the hit rate and $F$ indicates the false alarm rate. The decision bias was calculated using the formula $C = -(\Phi^{-1}(H) + \Phi^{-1}(F))/2$. The results are shown in Supplementary Figs. 2 and 3.

**Reporting summary**. Further information on research design is available in the Nature Research Reporting Summary linked to this article.

## Data availability
Behavioral data and fMRI decoding results have been deposited at OSF, accession code osf.io/zu7q9. Unthresholded fMRI activation maps have been deposited at NeuroVault (https://identifiers.org/neurovault.collection:6075).

## Code availability
Relevant codes have been deposited at OSF, accession code osf.io/zu7q9.

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

## Acknowledgements

This study was supported by a grant of the Deutsche Forschungsgemeinschaft (PO 548-14/2) and the European Fund for Regional Development (EFRE), ZS/2016/04/78113, Project: Center for Behavioral and Brain Sciences (CBBS). We thank Nico Marek for assisting with data collection.

## Author contributions

S.P., F.B., and L.W. developed the research question and experimental designs and interpreted the results. F.B., L.W., and F.R.K. programmed experimental code and acquired data. L.W., F.B., F.R.K., and M.H. analyzed data. S.P., F.B., and L.W. wrote the manuscript.

## Competing interests

The authors declare no competing interests.
