## [Peer Review File · Nature Communications]

Reviewers' comments:

Reviewer #1 (Remarks to the Author):

This manuscript describes a highly innovative and provocative set of fMRI experiments that test whether object-specific gaze patterns are represented in ventral occipital cortex. If true, the results reported here would strongly affect the way the field thinks about the ventral stream. I reviewed this paper several years ago, and it is clear that considerable effort and resources were invested in the revision, and in particular, the acquisition of additional experimental evidence to support their claim. However, strong claims require strong evidence, and I do not feel that the data presented here make an incontrovertible case that gaze patterns are represented in ventral occipital cortex.

In my initial review, I described a number of serious concerns, some of which have been addressed in a satisfactory manner, others of which have not, as described below.

1) Statistical evaluation of the decoding results. In my initial review, I asked that the authors perform non-parametric statistics on the decoding accuracies, since the authors had relied on tests that made assumptions about the underlying statistical distributions, which were most certainly not justified. I also asked that the authors ensure that the training and test sets were drawn from independent datasets. While the authors have complied with these recommendations, I now note idiosyncrasies in the data that are concerning. For example, in Figure 4, there are a number of data points that look to be significantly below chance. Why is this? The only way to obtain such a result is if there are some dependencies amongst the conditions. This is concerning. Also, the authors should display the 95th percentile of the null distribution for each plot, to show which data points are significant.

2) Along the same lines, the authors now state that they have modelled the imaging data with an HRF in all of the experiments. That is a big improvement, but unless I missed it, I do not see the time series data fit with the HRF model. What were the R² values of these model fits? I assume these model fits were used to derive the parameter estimates in Figure 3. What are the units on the y-axis in these plots? Typically, response amplitude is expressed in units of percent signal change.

3) Analyzing retinotopic visual cortex. In my initial review, I asked that the authors include analysis of V1-V3. The authors have now done so, but only in the most cursory fashion. Please show retinotopic maps with carefully delineated visual area boundaries. One general concern is that there are systematically different gaze patterns when viewing faces and houses. For example, subjects may dwell longer in the central part of the image during face viewing. If so, there may be differences between foveal and high eccentricity portions of V1. The authors could address this if they had performed proper retinotopic analysis on each participant.

4) The method used by the authors to control the variability of face and house paths is unclearly described and the effectiveness of the method was not demonstrated.

5) Figure 5b shows fixation distributions for two subjects (fw06 and ht02) viewing faces (blue) and houses (green). I find it highly unusual that several of the points for the two subjects appear to overlap perfectly. Are these really from two different subjects, or is there an error here? If the correlation between gaze locations is so strong between subjects, then I find it difficult to understand why it is impossible to classify across observer.

Reviewer #2 (Remarks to the Author):

This study by Wang and colleagues is very intriguing and presents results that could challenge how we think about the basis of representations in ventral temporal cortex. There is definitely much food for thought here, and the experiments have been well conducted and analysed. I was a reviewer on an earlier version of this manuscript. My primary concerns at that time were the differences in visual stimulation between the face and house eye track conditions and whether the generation of explicit eye movements was even necessary. In response to these concerns and related issues raised by the other reviewers, the authors have conducted a third experiment in which observers either followed a track of their own eye movements or someone else's. This is a very interesting manipulation that reveals decoding of face versus house tracks only for a subject's own pattern of eye movements.

The additional experiment indirectly addresses my concerns about the visual stimulation, under the assumption that the general tendencies for eye movements and the overall differences between face and house conditions will be similar for different subjects. Further, the authors suggest the data address the concern about the necessity of eye movements since the decoding sensitivity increases with the accuracy with which participants follow the fixation point.

These are somewhat unsatisfying responses, but I recognize the additional work the authors have done and think that the data should be published. Nevertheless, I have some suggestions for additional analyses and further comments that came to mind while reading through the new draft. The authors are making a strong claim and I think it is important they provide the strongest evidence possible.

Suggestions for Improvement

1) I repeat my suggestion for collecting additional data with central fixation. I think the question of whether eye movements are necessary is important particularly for the interpretation of the data. The difficulty with the argument that more accurate eye movements leads to more sensitive decoding is that it's not clear what accounts for the accuracy in the eye movements – presumably the subjects are not making different eye movements but might simply be blinking or just not attentive on those trials. If the authors do not want to collect data with central fixation, then I think they need to be more careful in their interpretation and acknowledge that they haven't tested the necessity of the eye movements.

2) While the additional experiment provides some evidence against the decoding reflecting differences in visual stimulation between the face and house eye track conditions, it also fails to replicate the findings from the first two experiments, since decoding there was based on other people's eye tracks. The authors don't really comment on this difference, but I think it needs to be addressed.

3) In general, throughout the manuscript there is very little information provided about the initial stimulus-driven eye movement sequences themselves. How many fixations were made per stimulus? What was the average saccade distance? How consistent were the movement sequences from stimulus to stimulus? Panel A in Figure 1 is not very informative. What exactly is plotted? Is it one subject's data? Across all possible stimuli? Characterizing the stimulus driven eye movement patterns is particularly important for the new experiment. How different were the eye movement patterns between subjects? In what ways did they differ? Did the difference in decoding for self- and other- tracks reflect the extent of difference between the eye movement patterns?

4) The correspondence between the gaze tracks and the elicited fixation sequences is also unclear. Figure 1C shows an example, but what exactly is depicted? One subject across 3 trials? How were these data selected for display and how representative are they? The gaze tracks for the different stimuli are compared with a classification approach, but the gaze tracks and the elicited eye movements in the dot tracking task are compared with a correlation based analysis. It would be better if the same approach were used to make both comparisons.

5) The analyses focus on specific ROIs, but I think it would be helpful to complement these results with a whole-brain searchlight analysis to reveal if there are other areas which also show a difference in responses between the different conditions.

6) The authors' interpretation seems to be that FFA/PPA may contain a motor program representation or that they analyze eye movement patterns. Isn't an alternative that FFA/PPA are associated with specific eye movement patterns and therefore those patterns are reflected in these areas without having any specific functional role? One key difference between natural viewing and dot following is that an entire motor program may be triggered by, for example, a face with a sequence of eye movements planned at stimulus onset. However, with dot following no such plan can be activated since the task is dot to dot and the subject has no idea of what the overall sequence will be.

Minor Comments

- 1) What was the task subjects were performing when the original eye movement data was collected for the first two experiments?
- 2) Were bilateral ROIs used? This is not clearly specified in the text.
- 3) Did the authors test for the difference between PPA/FFA and V1-V4. Or are they relying only on the difference in the significant effects in each case?
- 4) Do the authors have any idea why behavioral performance for dot detection is so much worse in experiment 3 compared to the other two experiments?
- 5) How large were the stimuli used to initially measure eye movements and how large was the display in the scanner? Were they matched?
- 6) Table 1 – what about V1-V4?
- 7) How variable were the house stimuli? Can the authors show more examples?

Reviewer #4 (Remarks to the Author):

NCOMMS

Individual face and house-related eye movement patterns distinctively activate FFA and PPA in the absence of faces and houses

Wang et al.

Summary

I was reviewer #4 on a previous version of this manuscript. In this revision, the authors have gone some way to address the concerns I raised in my original review.

The study presents a novel finding - that performing eye movements in a trajectory associated with different objects may be represented in the activity of different ventral stream areas. Initially, when reading and re-reading the manuscript, I thought that the authors were trying to make the "strong" version of the claim, i.e. that face-specific sequences are more distinguishable based on FFA activity and house-specific sequences are more distinguishable by PPA activity. But looking at the summary of decoding accuracies (Fig 4) that's not the case.

To me, taking all the multivariate analyses (and control conditions) together, the results suggest that there is a signal in FFA, PPA (but also FEF and SPL!) that support discrimination of the different face/object eye position trajectories. But it's still not completely clear from the results whether that's due to a stimulus, low-level related component in what is being picked up with fMRI. The authors have gone to some length to address those possibilities and I think the manuscript is written in such a way that it doesn't overstate the results.

The results of Experiment 3 - that a participant's own gaze patterns was distinguishable from another person's gaze pattern - based on the fMRI responses in FFA / PPA is interesting.

I second many of the detailed comments that the other reviewers have raised and although the classification accuracies are relatively low, they seem robust to the various controls that the authors were asked for. The overall idea of considering the responses in visual cortex and what fMRI measures in higher-level areas under slightly more ecologically valid conditions is a worthwhile goal.

We would like to thank the reviewers for their thoughtful comments. Please find our point-by-point responses below.

Reviewers' comments:

Reviewer #1 (Remarks to the Author):

This manuscript describes a highly innovative and provocative set of fMRI experiments that test whether object-specific gaze patterns are represented in ventral occipital cortex. If true, the results reported here would strongly affect the way the field thinks about the ventral stream. I reviewed this paper several years ago, and it is clear that considerable effort and resources were invested in the revision, and in particular, the acquisition of additional experimental evidence to support their claim. However, strong claims require strong evidence, and I do not feel that the data presented here make an incontrovertible case that gaze patterns are represented in ventral occipital cortex.

In my initial review, I described a number of serious concerns, some of which have been address in a satisfactory manor, others of which have not, as described below.

1) Statistical evaluation of the decoding results. In my initial review, I asked that the authors perform non-parametric statistics on the decoding accuracies, since the authors had relied on tests that made assumptions about the underlying statistical distributions, which were most certainly not justified. I also asked that the authors ensure that the training and test sets were drawn from independent datasets. While the authors have complied with these recommendations, I now note idiosyncrasies in the data that are concerning. For example, in Figure 4, there are a number of data points that look to be significantly below chance. Why is this? The only way to obtain such a result is if there are some dependencies amongst the conditions. This is concerning. Also, the authors should display the 95th percentile of the null distribution for each plot, to show which data points are significant.

-- While individual data points were below 50%, we did not observe significant below chance values in any of the contrast/ROI combinations in Fig. 4. Below-chance classification rate has been shown to be expected from neuroscience data, which are constrained to have small sample sizes, small effect sizes, but a large number of features (e.g., voxel numbers) (Jamalabadi et al., 2016). To address this issue, recommendations are: 1) a positive effect should be determined by statistical significance rather than the absolute value of the classification rate; 2) significance should be based on Monte Carlo methods rather than parametric tests; 3) twofold cross-validation, which usually leads to lower classification rates, is more suitable for hypothesis testing because of the smaller variance (Jamalabadi et al., 2016). Our data analysis protocol followed all of the three guidelines, which, to the best of our knowledge, ensures reliable results. Following your suggestions, we have now added the 95th percentile of the null distribution in the figure (Figure 4).

Reference

Jamalabadi H, Alizadeh S, Schönauer M, Leibold C, Gais S. (2016). Classification based hypothesis testing in neuroscience: below-chance level classification rates and overlooked statistical properties of linear parametric classifiers. *Human Brain Mapping*. 37:1842–1855. doi: 10.1002/hbm.23140.

2) Along the same lines, the authors now state that they have modelled the imaging data with an HRF in all of the experiments. That is a big improvement, but unless I missed it, I do not see the time series data fit with the HRF model. What were the R2 values of these model fits? I assume these model fits were used to derive the parameter estimates in Figure 3. What are the units on the y-axis in these plots? Typically, response amplitude is expressed in units of percent signal change.

-- We have now calculated the r-square values. For each voxel in each ROI, the design matrix of the general linear model (GLM) was multiplied by the parameter estimates (beta values) to generate the predicted time series (i.e., model fits). Then, the Pearson correlation coefficient between the predicted time series and the actual BOLD time series was calculated. The square of the correlation coefficient was obtained as the R square to show the variance of the BOLD signal explained by the model fitting. The table lists the mean R square averaged across all voxels in each ROI and averaged across all subjects. The corresponding standard errors are also listed.

	FFA		PPA		FEF		SPL	
	M	SE	M	SE	M	SE	M	SE
Exp. 1	0.06	4.6*10 ⁻⁴	0.06	3.4*10 ⁻⁴	0.07	4.0*10 ⁻⁴	0.09	4.7*10 ⁻⁴
Exp. 2	0.09	5.6*10 ⁻⁴	0.09	5.0*10 ⁻⁴	0.12	6.7*10 ⁻⁴	0.12	7.3*10 ⁻⁴
Exp. 3	0.05	6.4*10 ⁻⁴	0.03	4.0*10 ⁻⁴	0.06	5.2*10 ⁻⁴	0.05	2.9*10 ⁻⁴
	V1		V2		V3		V4	
	M	SE	M	SE	M	SE	M	SE
Exp. 1	0.14	7.0*10 ⁻⁴	0.12	7.1*10 ⁻⁴	0.09	6.5*10 ⁻⁴	0.09	7.4*10 ⁻⁴
Exp. 2	0.15	7.7*10 ⁻⁴	0.14	7.8*10 ⁻⁴	0.11	7.8*10 ⁻⁴	0.11	7.8*10 ⁻⁴
Exp. 3	0.06	7.4*10 ⁻⁴	0.05	7.6*10 ⁻⁴	0.05	8.2*10 ⁻⁴	0.05	9.1*10 ⁻⁴

The r-square values are in the range of other published r-square values of fMRI data (e.g. Kay et al, 2013, Fig. 6, doi.org/10.3389/fnins.2013.00247 or Kim and Maguire (2019), doi: 10.1016/j.neuroimage.2018.11.041, Suppl. Fig. 3). Kim and Maguire also show that r-square values in our range were sufficient to detect neural codes of abstract high-dimensional cognitive processes.

The y-axis in Figure 3 represents the beta values averaged across all the voxels in each ROI.

3) Analyzing retinotopic visual cortex. In my initial review, I asked that the authors include analysis of V1-V3. The authors have now done so, but only in the most cursory fashion. Please show retinotopic maps with carefully delineated visual area boundaries. One general concern is that there are systematically different gaze patterns when viewing faces and houses. For example, subjects may dwell longer in the central part of the image during face viewing. If so, there may be differences between foveal and high eccentricity portions of V1. The authors could address this if they had performed proper retinotopic analysis on each participant.

-- We did not carry out retinotopic mapping experiments initially for our subjects because our hypotheses were focused on the FFA and PPA (which we did map independently) but not on the early visual cortices. However, activation differences in central versus peripheral representation areas in early visual cortex should lead to above chance classification in our present analyses even though area borders (between V1, V2, V3, V4) were determined via probabilistic maps instead of retinotopic mapping of the individual brains. Thus, the results do not support these activation differences. Moreover, it would be difficult to run retinotopic mapping experiments on our participants, because many of them might not be at our university any longer.

4) The method used by the authors to control the variability of face and house paths is unclearly described and the effectiveness of the method was not demonstrated.

-- To control the spatial distribution of the fixation patterns, we firstly calculated the mean and the standard deviation of the Euclidean distances between the fixations for face pictures and the center of the screen. Then, we transformed the x, y coordinates of the fixations for house pictures to match the mean and standard deviation obtained from the face pictures. This procedure was conducted separately for each of the two subjects (see also methods, l. 490-501). We have now added the descriptions in the online methods and added a plot to show the fixation distribution after the normalization (Figure 1A). Furthermore, the mean dispersion for all stimuli is listed in Table S1.

5) Figure 5b shows fixation distributions for two subjects (fw06 and ht02) viewing faces (blue) and houses (green). I find it highly unusual that several of the points for the two subjects appear to overlap perfectly. Are these really from two different subjects, or is there an error here? If the correlation between gaze locations is so strong between subjects, then I find it difficult to understand why it is impossible to classify across observer.

-- We made a mistake when plotting the fixation distributions of the two subjects. The figure now has been corrected (see Figure 5). We thank you for notifying this point.

Reviewer #2 (Remarks to the Author):

This study by Wang and colleagues is very intriguing and presents results that could challenge how we think about the basis of representations in ventral temporal cortex. There is definitely much food for thought here, and the experiments have been well conducted and analysed. I was a reviewer on an earlier version of this manuscript. My primary concerns at that time were the differences in visual stimulation between the face and house eye track conditions and whether the generation of explicit eye movements was even necessary. In response to these concerns and related issues raised by the other reviewers, the authors have conducted a third experiment in which observers either followed a track of their own eye movements or someone else's. This is a very interesting manipulation that reveals decoding of face versus house tracks only for a subject's own pattern of eye movements.

The additional experiment indirectly addresses my concerns about the visual stimulation, under the assumption that the general tendencies for eye movements and the overall differences between face and house conditions will be similar for different subjects. Further, the authors suggest the data address the concern about the necessity of eye movements since

the decoding sensitivity increases with the accuracy with which participants follow the fixation point.

These are somewhat unsatisfying responses, but I recognize the additional work the authors have done and think that the data should be published. Nevertheless, I have some suggestions for additional analyses and further comments that came to mind while reading through the new draft. The authors are making a strong claim and I think it is important they provide the strongest evidence possible.

Suggestions for Improvement

1) I repeat my suggestion for collecting additional data with central fixation. I think the question of whether eye movements are necessary is important particularly for the interpretation of the data. The difficulty with the argument that more accurate eye movements leads to more sensitive decoding is that it's not clear what accounts for the accuracy in the eye movements – presumably the subjects are not making different eye movements but might simply be blinking or just not attentive on those trials. If the authors do not want to collect data with central fixation, then I think they need to be more careful in their interpretation and acknowledge that they haven't tested the necessity of the eye movements.

-- The eye blinks were detected by the eye-tracker and eliminated from data analysis using the python module cili (<https://github.com/beOn/cili>, see methods, l. 467-469). The quality of gaze following may, however, reflect fluctuations of vigilance, which is now discussed on l. 378-381. We also added a sentence that a replication with fixed fixation may be worthwhile.

2) While the additional experiment provides some evidence against the decoding reflecting differences in visual stimulation between the face and house eye track conditions, it also fails to replicate the findings from the first two experiments, since decoding there was based on other people's eye tracks. The authors don't really comment on this difference, but I think it needs to be addressed.

-- We intended to focus on the absolutely necessary eye movements for face recognition in Experiment 3. Therefore, we reduced the gaze tracks from 3000ms to 600ms. This of course led to less information that could be decoded. We think this is the reason we could only decode the self-generated - presumably optimal - sequences in FFA and PPA. On the other hand, this means that there is information in the gaze tracks beyond the initial 600ms that can be used to decode category-specific gaze tracks. This is now discussed on l. 407-414.

3) In general, throughout the manuscript there is very little information provided about the initial stimulus-driven eye movement sequences themselves. How many fixations were made per stimulus? What was the average saccade distance? How consistent were the movement sequences from stimulus to stimulus? Panel A in Figure 1 is not very informative. What exactly is plotted? Is it one subject's data? Across all possible stimuli? Characterizing the stimulus driven eye movement patterns is particularly important for the new experiment. How different were the eye movement patterns between subjects? In what ways did they differ? Did the difference in decoding for self- and other- tracks reflect the extent of difference between the eye movement patterns?

--The information of the stimulus-driven eye movements are shown in Table S1. To avoid redundancy, we reported the average saccadic amplitude, average number of saccades, average fixation duration and dispersion across all pictures for each condition (face and house). The average fixation numbers equal to the average number of saccades. The average saccade distance is shown in terms of the average dispersion of saccades. We have now added these characteristics from Experiment 3 to Table S1.

Figure 1A shows the fixation locations pooled from all pictures and from the two subjects. We added the information in the figure caption.

One aspect that contributed to the success of self-generated gaze track classification was the consistency of the first fixation location (l. 297-305). This was expected due to the stability of individual initial fixations - and their interindividual variability - that has been reported in the literature. This is discussed on l. 378-381

In response to your question on the eye movement patterns underlying the classification, we have now explored if the first-fixation distance between self-face and other-face, and the first-fixation distance between self-face and self-house, contribute to the decoding difference. The results showed that there was no correlation between the first-fixation distance and the decoding accuracy in either FFA (self-face vs. other-face: $r = 0.121$, $p = 0.665$; self-face vs. self-house: $r = 0.050$, $p = 0.853$) or PPA (self-face vs. other-face: $r = -0.374$, $p = 0.154$; self-face vs. self-house: $r = -0.368$, $p = 0.161$). Thus, it seems that the difference between gaze tracks is not simply reducible to the first fixation, but rather a more complex feature of the fixation sequences. This is in line with the finding discussed above, that information in the gaze tracks beyond the first 600ms appeared to contribute to the face vs. house discriminations in Experiments 1 and 2. In summary, to investigate the nature of the underlying gaze differences goes beyond the scope of this initial study. However, we agree that this must ultimately be the goal, now stated on l. 414.

4) The correspondence between the gaze tracks and the elicited fixation sequences is also unclear. Figure 1C shows an example, but what exactly is depicted? One subject across 3 trials? How were these data selected for display and how representative are they? The gaze tracks for the different stimuli are compared with a classification approach, but the gaze tracks and the elicited eye movements in the dot tracking task are compared with a correlation based analysis. It would be better if the same approach were used to make both comparisons.

-- In Figure 1C, the position (x, y coordinates, indicated by blue and green respectively) of the fixation dot and the actual eye position (x, y coordinates, indicated by red and turquoise respectively) of three consecutive trials from one subject are shown as a function of time. The x axis indicates the time (in seconds) relative to the onset of the first trial. Figure 1C is an example to show the correspondence between the gaze tracks and the elicited eye-movements. We did not intend to show "representative" trials but rather to show an intuitive example to the readers of how the correspondence was measured. We now have clarified this in the figure caption.

We used classification analysis for face- and house-related gaze tracks to infer if the activation in FFA and PPA contained the information about which category of stimuli had been presented. Comparing gaze tracks (i.e. dot sequences) with fixation sequences had a quite different purpose, namely to assess how closely the fixations followed the dots. Because

of these different goals, we feel that the use of classification and correlation analysis was appropriate. If we missed the purpose of your suggestion, please let us know.

5) The analyses focus on specific ROIs, but I think it would be helpful to complement these results with a whole-brain searchlight analysis to reveal if there are other areas which also show a difference in responses between the different conditions.

-- Following your suggestion, we ran a whole-brain search light analysis for the classification of self-face vs. self-house in Experiment 3, where FFA and PPA showed representational specificity of the gaze-tracks. Unfortunately, no cluster was obtained from the whole-brain search light under the conventional threshold ($p < 0.001$ at voxel-level, $p < 0.05$ at cluster level with FWE correction).

6) The authors' interpretation seems to be that FFA/PPA may contain a motor program representation or that they analyze eye movement patterns. Isn't an alternative that FFA/PPA are associated with specific eye movement patterns and therefore those patterns are reflected in these areas without having any specific functional role? One key difference between natural viewing and dot following is that an entire motor program may be triggered by, for example, a face with a sequence of eye movements planned at stimulus onset. However, with dot following no such plan can be activated since the task is dot to dot and the subject has no idea of what the overall sequence will be.

-- We fully agree that our experiment was quite artificial in that participants could not generate an eye movement pattern as they might when, e.g., viewing a face. We had discussed this on l. 420ff and suggested, that FFA and PPA may "analyze eye movement patterns - generated by parietal and frontal neurons - to help distinguishing faces from houses" to improve perception.

Due to the nature of fMRI data, it is correct that we cannot rule out that the activation patterns we observed have no functional implication. In essence, we would need to disturb FFA/PPA function to test its functional contribution. We now discuss the potential use of patient data to approach this issue (l. 428-436).

Minor Comments

1) What was the task subjects were performing when the original eye movement data was collected for the first two experiments?

-- In the first two experiments, subjects were freely looking at the pictures, each of which remained on the screen for 3 seconds (see methods, l. 462-464).

2) Were bilateral ROIs used? This is not clearly specified in the text.

-- All ROIs were bilateral. This has been specified in the text (l. 671).

3) Did the authors test for the difference between PPA/FFA and V1-V4. Or are they relying only on the difference in the significant effects in each case?

-- The classification accuracies for Face vs. House tracks were higher in FFA and PPA than in V1-V4 in both Experiment 1, $p < 0.001$ (permutation-based significance testing), and

Experiment 2, $p = 0.031$. In Experiment 3, the classification accuracies for self-generated gaze tracks (SF vs. SH) were higher in FFA and PPA than V1-V4, $p = 0.002$. These statistics have been added to the revised manuscript (see Results, l. 163-165, l. 203-205, and l. 273-275).

4) Do the authors have any idea why behavioral performance for dot detection is so much worse in experiment 3 compared to the other two experiments?

-- In Experiment 3, the moving dot was presented with the same speed and duration as the actual fixation patterns to maximize its similarity to the natural gaze patterns. This higher moving speed and shorter duration in contrast to the first experiments, which echoed by higher saccade amplitudes and shorter fixation durations (Table S1), made the black small dot even more difficult to detect (l. 250-253). Therefore, the detection performance was worse in Experiment 3 than in the other two experiments.

5) How large were the stimuli used to initially measure eye movements and how large was the display in the scanner? Were they matched?

-- The stimulus size was $10^\circ * 10^\circ$ of visual angle in Experiments 1 and 2 (l. 463), and was $18^\circ * 18^\circ$ of visual angle in Experiment 3 (l. 535). The sizes were matched for the lab session and the scanner session (l. 549-551).

6) Table 1 – what about V1-V4?

-- V1-V4 were defined by the probability maps in the Juelich Histological Atlas (Eickhoff et al., 2007, see l. 664). The cluster sizes in terms of voxel numbers are shown in Table 2.

7) How variable were the house stimuli? Can the authors show more examples?

-- Please see the figure below for the set of the house stimuli.

Reviewer #4 (Remarks to the Author):

NCOMMS

Individual face and house-related eye movement patterns distinctively activate FFA and PPA in the absence of faces and houses

Wang et al.

Summary

I was reviewer #4 on a previous version of this manuscript. In this revision, the authors have

gone some way to address the concerns I raised in my original review.

The study presents a novel finding - that performing eye movements in a trajectory associated with different objects may be represented in the activity of different ventral stream areas. Initially, when reading and re-reading the manuscript, I thought that the authors were trying to make the "strong" version of the claim, i.e. that face-specific sequences are more distinguishable based on FFA activity and house-specific sequences are more distinguishable by PPA activity. But looking at the summary of decoding accuracies (Fig 4) that's not the case.

To me, taking all the multivariate analyses (and control conditions) together, the results suggest that there is a signal in FFA, PPA (but also FEF and SPL!) that support discrimination of the different face/object eye position trajectories. But it's still not completely clear from the results whether that's due to a stimulus, low-level related component in what is being picked up with fMRI. The authors have gone to some length to address those possibilities and I think the manuscript is written in such a way that it doesn't overstate the results.

The results of Experiment 3 - that a participant's own gaze patterns were distinguishable from another person's gaze pattern - based on the fMRI responses in FFA / PPA is interesting.

I second many of the detailed comments that the other reviewers have raised and although the classification accuracies are relatively low, they seem robust to the various controls that the authors were asked for. The overall idea of considering the responses in visual cortex and what fMRI measures in higher-level areas under slightly more ecologically valid conditions is a worthwhile goal.

--Thank you for your positive comments!

Reviewers' comments:

Reviewer #1 (Remarks to the Author):

The authors have done an adequate job in responding to my review. I have no further comments.

Reviewer #2 (Remarks to the Author):

The authors have responded to the concerns raised in the previous round of review, although their responses do not always fully address the concerns. I remain supportive of the manuscript, but wish the authors would more explicitly acknowledge some of the limitations of the work.

I remain concerned that the authors have not completely ruled out the possible contribution of visual differences between the conditions and they have not demonstrated the necessity of the eye movements per se. I think the current data are sufficiently strong and interesting without explicit tests of these issues, but the authors should acknowledge these issues more directly and more substantially. For example, the authors use Experiment 3 to argue against general visual differences, assuming that overall differences between face and house conditions will be similar for different subjects. However, there is also a great reduction in the multivariate decoding results, which could reflect the difference in the stimulation periods (as the authors suggest) or it could suggest a contribution of visual factors to the results in Experiments 1 and 2. The authors are reluctant to collect additional data with central fixation (to show directly that eye movements are necessary), but fail to even acknowledge the question of the necessity of the eye movements. I strongly encourage the authors to more explicitly discuss these concerns and more generally to address the limitations of the study. Text could easily be added to the discussion that would strengthen the manuscript.

I suggested more analysis of the stimulus driven eye movement patterns, and although the authors have provided more data there are still additional analyses that would be interesting – for example characterizing the differences between individual subjects. In this light I want to encourage the authors to be more open with their data. Currently the authors state that data will be made available on request, but they could make at least some of it publicly available. While there may be additional challenges with the imaging data, the eye tracking data should be relatively easy to share, allowing others to investigate the eye tracking data more thoroughly.

The authors analyse the stimulus-driven eye movement sequences, the correspondence of fixations with dot presentation, and their general cross-correlation, but do not currently compare directly the actual eye movement patterns during dot fixation for the different conditions. Since the dot following is not perfect ($\rho < 0.7$), do the elicited eye movement patterns during dot fixation become more similar or perhaps more different for the different conditions (house versus face) relative to the stimulus driven fixations. This could be addressed using a similar analysis applied to the stimulus elicited fixation sequences (lines 127-132) and would be a useful analysis to include.

The authors applied a whole brain searchlight only in Experiment 3. What about the other two experiments? Further, even though no clusters were revealed for Experiment 3 at a conventional threshold, I think it would still be useful showing the analysis in supplementary material at a reduced threshold for exploratory purposes – possibly at a threshold where the FFA and PPA first appear, or even completely unthresholded.

Minor Comments

1) In the beginning of the results section, please state clearly that only two subjects provided stimulus driven eye movement data for Experiment 1 (line 117)

2) The alignment of columns in Table S2 seems off and the table is very hard to follow.

Reviewer #4 (Remarks to the Author):

NCOMMS

Individual face and house-related eye movement patterns distinctively activate FFA and PPA in the absence of faces and houses

Wang et al.

Summary

I was reviewer #4 on a previous version of this manuscript. In this revision, the authors have gone quite a long way to address the concerns other reviewers (and I) raised.

The study presents a novel finding - and in some respects the results are somewhat controversial (but in my opinion much less so, and less over-egged than many other imaging studies published in high-impact journals).

I agree with the other reviewers on testing some of the obvious potential confounds. (But I disagree about the level of burden of truth). In an ideal world, e.g. retinotopic maps (as suggested by another reviewer) and various other analyses could be added, but I am not sure this would necessarily help making the ultimate decision about this manuscript being published at NCOMMS or not (that's an editorial call).

Could this experiment have been done another way: the answer (as in so many other cases) is yes.

Have the author complied with reviewer requests and provided reasonable additional, apparently thoroughly performed analyses and data: yes.

There is a chance (also, "as in so many other cases") that there is an alternative interpretation about how the results arose, but it seems to me that what is presented here is v1.0 of a story that the authors should be allowed to present as a published paper and defend in public. An experiment to disprove the hypothesis presented here should be within reach of other labs (and the dialectic approach in science should be alive and kicking).

Reviewers' comments:

Reviewer #1 (Remarks to the Author):

The authors have done an adequate job in responding to my review. I have no further comments.

- We thank Reviewer 1 for the positive response.

Reviewer #2 (Remarks to the Author):

The authors have responded to the concerns raised in the previous round of review, although their responses do not always fully address the concerns. I remain supportive of the manuscript, but wish the authors would more explicitly acknowledge some of the limitations of the work.

I remain concerned that the authors have not completely ruled out the possible contribution of visual differences between the conditions and they have not demonstrated the necessity of the eye movements per se. I think the current data are sufficiently strong and interesting without explicit tests of these issues, but the authors should acknowledge these issues more directly and more substantially. For example, the authors use Experiment 3 to argue against general visual differences, assuming that overall differences between face and house conditions will be similar for different subjects. However, there is also a great reduction in the multivariate decoding results, which could reflect the difference in the stimulation periods (as the authors suggest) or it could suggest a contribution of visual factors to the results in Experiments 1 and 2. The authors are reluctant to collect additional data with central fixation (to show directly that eye movements are necessary), but fail to even acknowledge the question of the necessity of the eye movements. I strongly encourage the authors to more explicitly discuss these concerns and more generally to address the limitations of the study. Text could easily be added to the discussion that would strengthen the manuscript.

- We had discussed the potential contribution of visual factors in some detail in the discussion to explain our design choices for Experiment 3 (now l. 354-369). Also, in the last revision, we had mentioned the problem of eye movements in the discussion (l. 376-381 in r2, now l. 382-390) We have now added an explicit discussion of how a potential experiment with central fixation requirement may potentially strengthen the data (l. 391-395). However, we also discuss our previously mentioned concern that in spite of central fixation, it might be difficult to control microsaccades. Nevertheless, the role of microsaccades may in itself be an interesting future topic. Moreover, we explicitly state in the discussion that visual stimulation differences may have contributed to the higher classification accuracies in Exp. 1 and 2 (l. 428-430).

I suggested more analysis of the stimulus driven eye movement patterns, and although the authors have provided more data there are still additional analyses that would be interesting – for example characterizing the differences between individual subjects. In this light I want to encourage the authors to be more open with their data. Currently the authors state that data will be made available on request, but they could make at least some of it publicly available. While there may be additional challenges with the imaging data, the eye tracking data should

be relatively easy to share, allowing others to investigate the eye tracking data more thoroughly.

- We agree that data should be shared. We will make the data available via OSF. (We have already started this process, see <https://osf.io/zu7q9/>). However, because both the first author (Dr. Wang) and I are currently out of office, this process may need some more time to complete. However, we will provide all data until potential publication.

The authors analyse the stimulus-driven eye movement sequences, the correspondence of fixations with dot presentation, and their general cross-correlation, but do not currently compare directly the actual eye movement patterns during dot fixation for the different conditions. Since the dot following is not perfect ($\rho < 0.7$), do the elicited eye movement patterns during dot fixation become more similar or perhaps more different for the different conditions (house versus face) relative to the stimulus driven fixations. This could be addressed using a similar analysis applied to the stimulus elicited fixation sequences (lines 127-132) and would be a useful analysis to include.

- We have now included the requested analysis on l. 282-291: The classification analysis on the actual eye-movement data (x and y coordinates) while subjects were following the gaze patterns showed a mean accuracy of 57.0% (SE = 1.4%) for self-face (SF) vs. self-house (SH), and a mean accuracy of 54.5% (SE = 1.7%) for other-face (OF) vs. other-house (OH), a mean accuracy of 63.3% (SE = 2.7%) for SF vs. OF, and a mean accuracy of 60.3% (SE = 1.7%) for SH vs. OH. The eye-movement patterns in distinguishing OF vs. OH, SF vs. OF, SH vs. OH dot patterns did not differ from the eye-movement patterns in distinguishing SF vs. SH dot patterns, $p = 0.157$ (permutation-based significance testing). Therefore, the FFA- and PPA-specific classification of SF vs. SH cannot be reduced to the physical properties of the stimulus-driven eye-movement data.

The authors applied a whole brain searchlight only in Experiment 3. What about the other two experiments? Further, even though no clusters were revealed for Experiment 3 at a conventional threshold, I think it would still be useful showing the analysis in supplementary material at a reduced threshold for exploratory purposes – possibly at a threshold where the FFA and PPA first appear, or even completely unthresholded.

- We have now also calculated searchlight analyses for Experiments 1 and 2. The results mirror the searchlight result for Experiment 3 in that no significant clusters were obtained (results added to supplementary data on l. 1116-1121). Given the problems with too liberal thresholds that led to non-replicable imaging results in the past (e.g. Eklund et al., PNAS 2016), we think it would be inappropriate to lower the threshold of our searchlight analyses. Particularly, given that we present data on a very novel hypothesis, we want to make sure that the data we present are reliable. However, we have added the searchlight results for all three experiments, thresholded at a liberal threshold ($p < 0.005$ at voxel-level, uncorrected at cluster level), for your inspection.

Minor Comments

1) In the beginning of the results section, please state clearly that only two subjects provided stimulus driven eye movement data for Experiment 1 (line 117)

- This statement has now been added.

2) The alignment of columns in Table S2 seems off and the table is very hard to follow.

- We have changed Table S1 (S2 does not exist) to improve readability.

Reviewer #4 (Remarks to the Author):

NCOMMS

Individual face and house-related eye movement patterns distinctively activate FFA and PPA in the absence of faces and houses

Wang et al.

Summary

I was reviewer #4 on a previous version of this manuscript. In this revision, the authors have gone quite a long way to address the concerns other reviewers (and I) raised.

The study presents a novel finding - and in some respects the results are somewhat controversial (but in my opinion much less so, and less over-egged than many other imaging studies published in high-impact journals).

I agree with the other reviewers on testing some of the obvious potential confounds. (But I disagree about the level of burden of truth). In an ideal world, e.g. retinotopic maps (as suggested by another reviewer) and various other analyses could be added, but I am not sure this would necessarily help making the ultimate decision about this manuscript being published at NCOMMS or not (that's an editorial call).

Could this experiment have been done another way: the answer (as in so many other cases) is yes.

Have the author complied with reviewer requests and provided reasonable additional, apparently thoroughly performed analyses and data: yes.

There is a chance (also, "as in so many other cases") that there is an alternative interpretation about how the results arose, but it seems to me that what is presented here is v1.0 of a story that the authors should be allowed to present as a published paper and defend in public. An experiment to disprove the hypothesis presented here should be within reach of other labs (and the dialectic approach in science should be alive and kicking).

- We thank the reviewer for the positive comments.

REVIEWERS' COMMENTS:

Reviewer #2 (Remarks to the Author):

The authors responses are satisfactory and I have no additional coments. This is a provocative manuscript with results that should encourage deeper investigation of the issues raised.

REVIEWERS' COMMENTS:

Reviewer #2 (Remarks to the Author): The authors responses are satisfactory and I have no additional comments. This is a provocative manuscript with results that should encourage deeper investigation of the issues raised.

- We thank the reviewer for the assessment and we agree that deeper investigations should follow.